# Key features of the genetic architecture and evolution of host-microbe interactions revealed by high-resolution genetic mapping of the mucosa-associated gut microbiome in hybrid mice

**Shauni Doms**[1,2], **Hanna Fokt**[1,2], **Malte Christoph Rühlemann**[3,4], **Cecilia J Chung**[1,2], **Axel Kuenstner**[5], **Saleh M Ibrahim**[5,6], **Andre Franke**[3], **Leslie M Turner**[7]*[†], **John F Baines**[1,2]*[†]

[1]Max Planck Institute for Evolutionary Biology, Plön, Germany; [2]Section of Evolutionary Medicine, Institute for Experimental Medicine, Kiel University, Kiel, Germany; [3]Institute for Clinical Molecular Biology (IKMB), Kiel University, Kiel, Germany; [4]Institute for Medical Microbiology and Hospital Epidemiology, Hannover Medical School, Hannover, Germany; [5]Institute of Experimental Dermatology, University of Lübeck, Lübeck, Germany; [6]Sharjah Institute of Medical Research, Sharjah, United Arab Emirates; [7]Milner Centre for Evolution, Department of Biology & Biochemistry, University of Bath, Bath, United Kingdom

*For correspondence:
l.m.turner@bath.ac.uk (LMT);
Baines@evolbio.mpg.de (JFB)

†These authors contributed
equally to this work

Competing interest: The authors
declare that no competing
interests exist.

Reviewing Editor: Jenny Tung,
Duke University, United States

**Abstract:** Determining the forces that shape diversity in host-associated bacterial communities is critical to understanding the evolution and maintenance of metaorganisms. To gain deeper understanding of the role of host genetics in shaping gut microbial traits, we employed a powerful genetic mapping approach using inbred lines derived from the hybrid zone of two incipient house mouse species. Furthermore, we uniquely performed our analysis on microbial traits measured at the gut mucosal interface, which is in more direct contact with host cells and the immune system. Several mucosa-associated bacterial taxa have high heritability estimates, and interestingly, 16S rRNA transcript-based heritability estimates are positively correlated with cospeciation rate estimates. Genome-wide association mapping identifies 428 loci influencing 120 taxa, with narrow genomic intervals pinpointing promising candidate genes and pathways. Importantly, we identified an enrichment of candidate genes associated with several human diseases, including inflammatory bowel disease, and functional categories including innate immunity and G-protein-coupled receptors. These results highlight key features of the genetic architecture of mammalian host-microbe interactions and how they diverge as new species form.

## Editor's evaluation

This paper uses inbred hybrid mouse lines to estimate the heritability of the mucosa-associated microbiome and map variants in the mouse genome that are associated with the composition of the microbiome. The findings are of broad interest to microbiome researchers and improve on knowledge in the field, as the mapping design facilitates the identification of narrow association intervals and points to a novel correlation between heritability and cospeciation rates. The manuscript provides useful information about the approach to heritability estimation, allowing the results to be more readily placed in context. Congratulations on this important contribution to the literature.

**eLife digest** The digestive system, particularly the large intestine, hosts many types of bacteria which together form the gut microbiome. The exact makeup of different bacterial species is specific to an individual, but microbiomes are often more similar between related individuals, and more generally, across related species. Whether this is because individuals share similar environments or similar genetic backgrounds remains unclear. These two factors can be disentangled by breeding different animal lineages – which have different genetic backgrounds while belonging to the same species – and then raising the progeny in the same environment.

To investigate this question, Doms et al. studied the genes and microbiomes of mice resulting from breeding strains from multiple locations in a natural hybrid zone between different subspecies. The experiments showed that 428 genetic regions affected the makeup of the microbiome, many of which were known to be associated with human diseases. Further analysis revealed 79 genes that were particularly interesting, as they were involved in recognition and communication with bacteria. These results show how the influence of the host genome on microbiome composition becomes more specialized as animals evolve.

Overall, the work by Doms et al. helps to pinpoint the genes that impact the microbiome; this knowledge could be helpful to examine how these interactions contribute to the emergence of conditions such as diabetes or inflammatory bowel disease, which are linked to perturbations in gut bacteria.

## Introduction

The recent widespread recognition of the gut microbiome's importance to host health and fitness represents a critical advancement of biomedicine. Host phenotypes affected by the gut microbiome are documented in humans (*Ley et al., 2006*; *Turnbaugh et al., 2009*; *Lynch and Pedersen, 2016*), laboratory animals (*Bäckhed et al., 2004*; *Turnbaugh et al., 2008*; *Rolig et al., 2015*; *Rosshart et al., 2017*; *Gould et al., 2018*), and wild populations (*Suzuki, 2017*; *Roth et al., 2019*; *Suzuki et al., 2020a*; *Hua et al., 2020*), and include critical traits such as aiding digestion and energy uptake (*Rowland et al., 2018*), and the development and regulation of the immune system (*Davenport, 2020*).

Despite the importance of the gut microbiome, community composition varies significantly among host species, populations, and individuals (*Benson et al., 2010*; *Yatsunenko et al., 2012*; *Brooks et al., 2016*; *Rehman et al., 2016*; *Amato et al., 2019*). While a portion of this variation is expected to be selectively neutral, alterations of the gut microbiome are on the one hand linked to numerous human diseases (*Carding et al., 2015*; *Lynch and Pedersen, 2016*), including diabetes (*Qin, 2012*), inflammatory bowel disease (IBD) (*Ott et al., 2004*; *Gevers et al., 2014*), and mental disorders (*Clapp et al., 2017*). On the other hand, there is evidence that the gut microbiome can play an important role in adaptation on both recent (*Hehemann et al., 2010*; *Suzuki and Ley, 2020b*) and ancient evolutionary timescales (*Rausch et al., 2019*). Collectively, these phenomena suggest that it would be evolutionarily advantageous for hosts to influence their microbiome.

An intriguing observation made in comparative microbiome research in the last decade is that the pattern of diversification among gut microbiomes appears to mirror host phylogeny (*Ochman et al., 2010*). This phenomenon, coined 'phylosymbiosis' (*Brucker and Bordenstein, 2012a*; *Brucker and Bordenstein, 2012b*; *Lim and Bordenstein, 2020*), is documented in a number of diverse host taxa (*Brooks et al., 2016*) and also extends to the level of the phageome (*Gogarten et al., 2021*). Several non-mutually exclusive hypotheses are proposed to explain phylosymbiosis (*Moran and Sloan, 2015*). However, it is likely that vertical inheritance is important for at least a subset of taxa, as signatures of cospeciation/codiversification are present among numerous mammalian associated gut microbes (*Moeller et al., 2016*; *Groussin et al., 2017*; *Moeller et al., 2019*), which could also set the stage for potential coevolutionary processes. Importantly, experiments involving interspecific fecal microbiota transplants indeed provide evidence of host adaptation to their conspecific microbial communities (*Brooks et al., 2016*; *Moeller et al., 2019*). Furthermore, cospeciating taxa were observed to be significantly enriched among the bacterial species depleted in early onset IBD, an immune-related disorder, suggesting a greater evolved dependency on such taxa (*Papa et al., 2012*; *Groussin et al.,*

*2017*). However, the nature of genetic changes involving host-microbe interactions that take place as new host species diverge remains underexplored.

House mice are an excellent model system for evolutionary microbiome research, as studies of both natural populations and laboratory experiments are possible (*Suzuki, 2017*; *Suzuki et al., 2019*). In particular, the house mouse species complex is composed of subspecies that hybridize in nature, enabling the potential early stages of codiversification to be studied. We previously analysed the gut microbiome across the Central European hybrid zone of *Mus musculus musculus* and *Mus musculus domesticus* (*Wang et al., 2015*), which share a common ancestor ~0.5 million years ago (*Geraldes et al., 2008*). Importantly, transgressive phenotypes (i.e. exceeding or falling short of parental values) among gut microbial traits as well as increased intestinal histopathology scores were common in hybrids, suggesting that the genetic basis of host control over microbes has diverged (*Wang et al., 2015*). The same study performed an $F_2$ cross between wild-derived inbred strains of *M. m. domesticus* and *M. m. musculus* and identified 14 quantitative trait loci (QTL) influencing 29 microbial traits. However, like classical laboratory mice, these strains had a history of rederivation and reconstitution of their gut microbiome, thus leading to deviations from the native microbial populations found in nature (*Rosshart et al., 2017*; *Org and Lusis, 2018*), and the genomic intervals were too large to identify individual genes.

In this study, we employed a powerful genetic mapping approach using inbred lines directly derived from the *M. m. musculus*–*M. m. domesticus* hybrid zone, and further focus on the mucosa-associated microbiota due to its more direct interaction with host cells (*Fukata and Arditi, 2013*; *Chu and Mazmanian, 2013*), distinct functions compared to the luminal microbiota (*Wang et al., 2010*; *Vaga et al., 2020*), and greater dependence on host genetics (*Spor et al., 2011*; *Linnenbrink et al., 2013*). Previous mapping studies using hybrids raised in a laboratory environment showed that high mapping resolution is possible due to the hundreds of generations of natural admixture between parental genomes in the hybrid zone (*Turner and Harr, 2014*; *Pallares et al., 2014*; *Škrabar et al., 2018*). Accordingly, we here identify 428 loci contributing to variation in 120 taxa, whose narrow genomic intervals (median <2 Mb) enable many individual candidate genes and pathways to be pinpointed. We identify a high proportion of bacterial taxa with significant heritability estimates and find that bacterial phenotyping based on 16S rRNA transcript compared to gene copy-based profiling yields an even higher proportion. Furthermore, these heritability estimates also significantly positively correlate with cospeciation rate estimates, suggesting a more extensive host genetic architecture for cospeciating taxa. Finally, we identify numerous enriched functional pathways, whose role in host-microbe interactions may be particularly important as new species form.

## Results
### Microbial community composition

To obtain microbial traits for genetic mapping in the G2 mapping population, we sequenced the 16S rRNA gene from caecal mucosa samples of 320 hybrid male mice based on DNA and RNA (cDNA), which reflect bacterial cell number and activity, respectively. After applying quality filtering and subsampling 10,000 reads per sample, we identified a total of 4684 amplicon sequence variants (ASVs). For further analyses, we established a 'core microbiome' (defined in Materials and methods), such that analyses were limited to those taxa common and abundant enough to reveal potential genetic signal. The core microbiome is composed of four phyla, five classes, five orders, 11 families, 27 genera, and 90 ASVs for RNA, and four phyla, five classes, six orders, 12 families, 28 genera and 46 ASVs for DNA. A combined total of 98 unique ASVs belong to the core, of which 38 were shared between DNA and RNA (*Figure 1—figure supplement 1*). The most abundant genus in our core microbiome is *Helicobacter* (*Figure 1—figure supplement 2B*), consistent with a previous study of the wild hybrid *M. m. musculus/M. m. domesticus* mucosa-associated microbiome (*Wang et al., 2015*).

Importantly, inspection of the taxonomic profiles of the mapping population confirms that key features of the native mouse microbiome were retained in our wild-derived lines, despite multiple generations of breeding in the laboratory. For example, the number, identity, and relative proportions of major bacterial orders are more similar to wild-caught mice than to rederived classical laboratory strains (*Figure 1—figure supplement 2A* – Order-level bar plot; Fig S4 from *Rosshart et al., 2017*). This is consistent with a previous study which showed that microbiomes of wild-derived strains

maintained their distinctiveness over 10 generations of breeding in the laboratory (*Moeller et al., 2018*).

## Heritability

Next, we estimated the narrow-sense heritability ($h^2$) of bacterial traits using lme4QTL (*Ziyatdinov et al., 2018*). Of the 153 total core taxa, we identified 21 taxa for DNA and 30 taxa for RNA with significant heritability estimates ($P_{RLRT}$<0.05, Restricted Likelihood Ratio test), with estimates ranging between 39 and 83% (*Figure 1A–B* and *Supplementary file 1*). The top values for bacterial abundances are similar to heritability estimates for body weight (87%) and body length (67%). ASV97 (genus *Oscillibacter*) followed by the genus *Paraprevotella,* and ASV7 (genus *Paraprevotella*) showed the highest heritability among DNA-based traits (80.8, 78.6, and 77.4%, respectively; *Figure 1A*), while ASV97 (genus *Oscillibacter*), followed by ASV36 (genus *Oscillibacter*), and ASV135 (unclassified order Bacteroidales) had the highest heritability among RNA-based traits (83.0, 80.2, and 79.3%, respectively; *Figure 1B*). The heritability estimates for DNA- and RNA-based measurements of the same taxa are significantly correlated (Spearman's rho = 0.53, p=3.1 × 10$^{-12}$, *Figure 1—figure supplement 3*).

Next, we estimated the 'chip' heritability (CH), i.e., the percentage of phenotypic variance in bacterial traits explained by genotyped SNPs (32,625 SNPs; *Zhou et al., 2013*). We find 23 DNA-based traits and 27 RNA-based traits with significant chip heritability estimates, ranging between 50.0 and 15.9% (*Figure 1A, B* and *Supplementary file 1*).

We compared these heritability estimates to estimates from previous studies in other mammals (*Supplementary file 1*), including mice (*O'Connor et al., 2014*; *Org et al., 2015*), humans (*Davenport et al., 2015*; *Goodrich et al., 2016*; *Turpin et al., 2016*; *Xu et al., 2020*; *Ishida et al., 2020*; *Hughes et al., 2020*; *Kurilshikov et al., 2021*), pigs (*Chen et al., 2018*), and primates (*Grieneisen et al., 2021*). DNA-based heritability estimates are positively correlated with DNA-based heritability estimates from male mice (Spearman's rho = 0.60, p=0.049, n=11; *Org et al., 2015*; *Figure 1—figure supplement 5A*) and with DNA-based heritability estimates from one human study (Spearman's rho = 0.38, p=0.049, n=28; *Turpin et al., 2016*; *Figure 1—figure supplement 5B*).

## Heritability estimates are correlated with predicted cospeciation rates

In an important meta-analysis of the gut microbiome across diverse mammalian taxa, *Groussin et al., 2017* estimated cospeciation rates of individual bacterial taxa by measuring the congruence of host and bacteria phylogenetic trees relative to the number of host-swap events. We reasoned that taxa with higher cospeciation rates might also demonstrate higher heritability, as these more intimate evolutionary relationships would provide a greater opportunity for genetic aspects to evolve. Intriguingly, we observe a significant positive correlation for RNA-based traits ($h^2$, p=0.037, Spearman's rho = 0.47; CH, p=0.012, Spearman's rho = 0.55; *Figure 1D*; *Figure 1—figure supplement 4D*), but not for DNA-based traits ($h^2$, Spearman's rho = −0.062, p=0.80; CH, Spearman's rho = −0.091, p=0.70; *Figure 1C*; *Figure 1—figure supplement 4C*) for both narrow-sense heritability and chip heritability estimates. To evaluate whether these results may be confounded by taxon abundance, we further used a multiple linear regression model incorporating both taxon abundance and cospeciation rate. The overall regression model was not significant ($R^2$=0.27, $F_{(2,17)}$=3.202, p=0.067). Furthermore, the cospeciation rate significantly predicts the heritability estimate (p=0.022), while the median abundance does not (p=0.92). Thus, heritability estimates and cospeciation rates are associated independent of taxon abundance. These results support the notion that cospeciating taxa evolved a greater dependency on host genes, and further suggest that bacterial activity may better reflect the underlying biological interactions.

## Genetic mapping of host loci determining microbiome composition

Next, we performed genome-wide association mapping of the relative abundances of core taxa, in addition to two alpha-diversity measures (Shannon and Chao1 indices), based on 32,625 SNPs. We used a linear mixed model including both additive and dominance terms in the model to enable the identification of underdominance and overdominance in this hybrid mapping population. We included mating pair and the genotype-based genomic relatedness matrix (GRM) as random effects to control for maternal effects and relatedness, respectively (see Materials and methods). While we found no

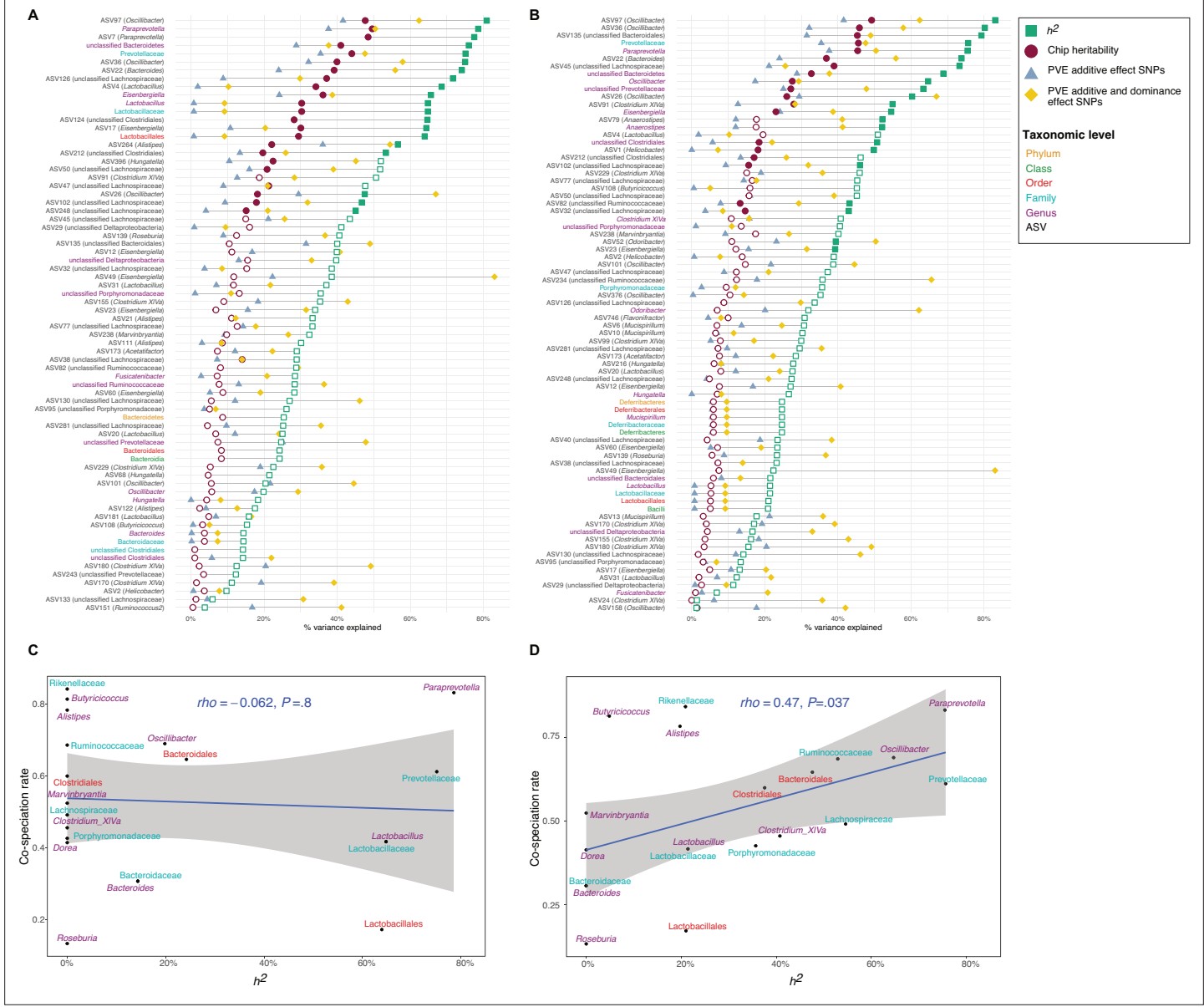

**Figure 1.** Heritability estimates and their relationship to cospeciation rates.

(A–B) Lollipop chart showing the values of the chip heritability (dark red circles), narrow-sense heritability (green squares), PVE by the additive effect (blue triangles), and additive and dominance effect (yellow diamonds) of all significant single nucleotide polymorphisms (SNPs) within one taxon for DNA-based traits (A) and RNA-based traits (B) Only taxa with a non-zero narrow-sense heritability estimate are shown. Only the outline of non-significant heritability estimates (p<0.05, Restricted Likelihood Ratio test) are shown. Taxa without significant hits have no PVE reported. (C–D) Relationship between the narrow-sense heritability estimates for the relative abundance of bacterial taxa and cospeciation rate for the same genus calculated by *Groussin et al., 2017*. DNA level (C), and RNA level (D). The blue line represents a linear regression fit to the data and the grey area the corresponding confidence interval. p-Values and the correlation coefficient are calculated using the Spearman's correlation test. The text labels on the y-axis (A–B) and the text labels in panels (C-D) are coloured according to taxonomic level: amplicon sequence variants (ASV) in black, genus in purple, family in light blue, order in red, class in green, and phylum in yellow.

The online version of this article includes the following figure supplement(s) for figure 1:

**Figure supplement 1.** Selection of taxa for mGWAS analysis.

**Figure supplement 2.** Relative abundances of core taxa.

**Figure supplement 3.** Correlation of single nucleotide polymorphisms (SNP)-based heritability estimates based on DNA (x-axis) or RNA (y-axis).

*Figure 1 continued on next page*

Figure 1 continued

**Figure supplement 4.** Relationship between the chip heritability estimates for the relative abundance of bacterial taxa and cospeciation rate for the same genus calculated by *Groussin et al., 2017*.

**Figure supplement 5.** Correlation between heritability estimates.

**Figure supplement 6.** Overview of the intercross design.

genome-wide significant associations for alpha diversity at either the DNA or RNA level (p>1.53 × $10^{-6}$), a total of 1030 genome-wide significant associations were identified for individual taxa (p<1.53 × $10^{-6}$, *Supplementary file 2*), of which 428 achieved study-wide significance (p<1.29 × $10^{-8}$). Apart from the X chromosome, all autosomal chromosomes contained study-wide significant associations

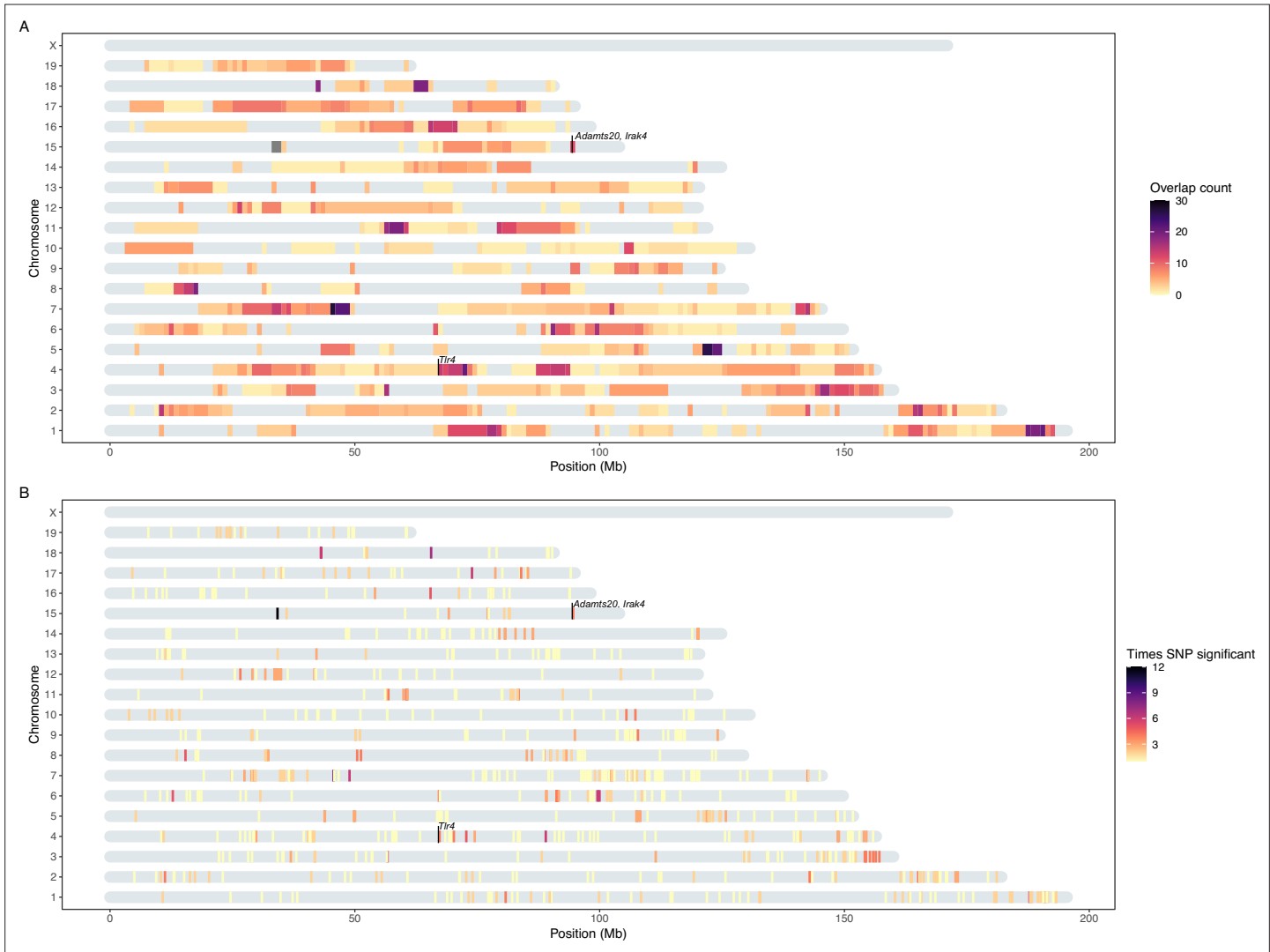

**Figure 2.** Heatmap of significant host loci from association mapping of bacterial abundances. Karotype plot showing the number of significant loci found using a study-wide threshold, where (**A**) plots the significance intervals and (**B**) the significant single nucleotide polymorphisms (SNP) markers on the chromosomes. The position of the SNPs on panel (B) has been amplified by 0.5 Mb to visualise it. The position of the genes closest to SNPs with the lowest p-values (*Tlr4*, and *Irak4* and *Adamsts20*) are indicated.

The online version of this article includes the following figure supplement(s) for figure 2:

**Figure supplement 1.** Manhattan plots for ASV184 (*Dorea).*

**Figure supplement 2.** Number of significantly associated loci per bacterial taxon.

**Table 1.** Overview of mapping statistics.

Loci with a P-value below the study-wide threshold (P<1.29 × 10⁻⁸) are considered significant. 'Significant loci total P' are the loci having a significant P-value from the total model (additive and dominance effect), 'Significant loci additive P' have a significant additive effect, and 'Significant loci dominance P' have a significant dominance effect.

|  | DNA | RNA | Total |
|---|---|---|---|
| Mapped taxa | 101 | 142 | 153 |
| Taxa with significant loci | 65 | 93 | 120 |
| Median interval size (Mb) | 1.32 | 2.5 | 1.91 |
| Total significant SNPs | 316 | 596 | 782 |
| Total significant loci | 443 | 746 | 1184 |
| Unique significant loci | 172 | 305 | 428 |
| Significant loci total P | 83 | 144 | 204 |
| Significant loci additive P | 144 | 244 | 351 |
| Significant loci dominance P | 88 | 171 | 230 |
| Median significant loci per trait | 5 | 5 | 8 |
| Median unique significant loci per trait | 3 | 3 | 4 |
| Median unique significant SNPs per locus | 2 | 2 | 2 |
| Median number of genes per locus | 32 | 54 | 43 |
| Median protein coding genes per locus | 11 | 17 | 14 |

SNPs: single nucleotide polymorphisms.

(*Figure 2*). Out of the 153 mapped taxa, 120 had at least one significant association (*Table 1*). For the remainder of our analyses, we focus on the results using the more stringent study-wide threshold, and combined significant SNPs within 10 Mb into significant regions (*Supplementary file 3*). The median size of significant regions is 1.91 Mb, which harbour a median of 14 protein-coding genes. On average, we observe five significant mouse genomic regions per bacterial taxon.

Of the significant loci with estimated interval sizes, we find 69 intervals (16.1%) that are smaller than 1 Mb (*Supplementary file 4*). The smallest interval is only 231 bases and associated with the RNA-based abundance of an unclassified genus belonging to Deltaproteobacteria. It is situated in an intron of the C3 gene, a complement component playing a central role in the activation of the complement system, which modulates inflammation and contributes to antimicrobial activity (*Ricklin et al., 2016*).

The significant genomic regions and SNPs are displayed in *Figure 2A and B*, respectively. Individual SNPs were associated with up to 14 taxa, and significant intervals with up to 30 taxa. The SNPs with the lowest p-values were associated with the genus *Dorea* and two ASVs belonging to *Dorea* (ASV184 and ASV293; *Figure 2—figure supplement 1*). At the RNA level this involves two loci: mm10-chr4: 67.07 Mb, where the peak SNP is 13 kb downstream of the closest gene *Tlr4* (UNC7414459, p=2.31 × 10⁻⁶⁹, additive p=4.48 × 10⁻¹¹⁸, dominance p=1.37 × 10⁻¹¹¹; *Figure 2*; *Figure 2—figure supplement 1*), and mm10-chr15: 94.4 Mb, where the peak SNP is found within the *Adamts20* gene (UNC26145702, p=4.51 × 10⁻⁶⁵, additive p=1.87 × 10⁻¹¹³, dominance p=1.56 × 10⁻¹⁰⁵; *Figure 2*; *Figure 2—figure supplement 1*). Interestingly, the *Irak4* gene, whose protein product is rapidly recruited after TLR4 activation, is also located 181 kb upstream of *Adamts20*. The five taxa displaying the most associations were ASV19 (*Bacteroides*), *Dorea*, ASV36 (*Oscillibacter*), ASV35 (*Bacteroides*), and ASV98 (unclassified Lachnospiraceae) (*Figure 2—figure supplement 2*).

## Ancestry, dominance, and effect sizes

A total of 398 significant SNPs were ancestry informative between *M. m. musculus* and *M. m. domesticus* (i.e. represent fixed differences between subspecies). To gain further insight into the genetic architecture of microbial trait abundances, we estimated the degree of dominance at each significant

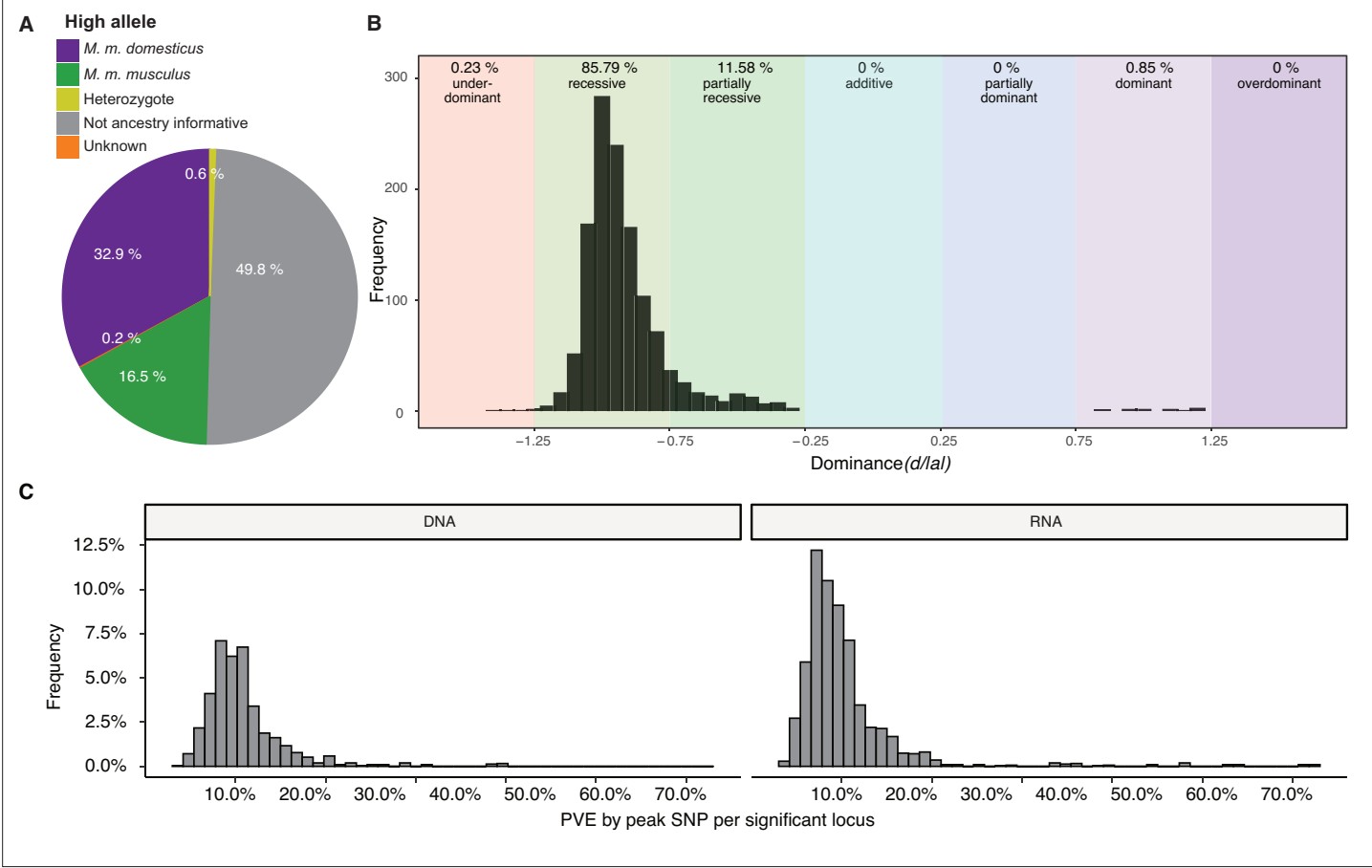

**Figure 3.** Genetic architecture of significant loci. (**A**) Source of the allele with the highest phenotypic value. (**B**) Histogram of dominance values d/a of significant loci reveals a majority of loci acting recessive or partially recessive. (**C**) Histogram showing the percentage of variance explained (PVE) by the peak single nucleotide polymorphisms (SNP) for DNA (left) and RNA (right).

locus using the *d/a* ratio (*Falconer, 1996*), where alleles with strictly recessive, additive, and dominant effects have *d/a* values of –1, 0, and 1, respectively. As half of the SNPs were not ancestry informative (*Figure 3A*), it was not possible to consistently have *a* associated with one parent/subspecies, hence we report *d/|a|* such that it can be interpreted with respect to bacterial abundance. For the vast majority of loci (83.79%), the allele associated with higher abundance is recessive or partially recessive (–1.25<*d/|a|*<–0.75; *Figure 3B*). On the basis of the arbitrary cutoffs, we used to classify dominance, only a small proportion of alleles are underdominant (0.23%; *d/|a|*<–1.25). However, for one-third of the significant SNPs, the heterozygotes display transgressive phenotypes, i.e., mean abundances that are either significantly lower (31% of SNPs) or higher (2% of SNPs) than those of both homozygous genotypes. Interestingly, the *domesticus* allele was associated with higher bacterial abundance in two-thirds of this subset (33.9 vs. 16.5% *musculus* allele; *Figure 3A*).

Next, we estimated phenotypic effect sizes by calculating the percent variance explained (PVE) by the peak SNP of each significant region. Peak SNPs explain between 3 and 64% of the variance in bacterial abundance, with a median effect size of 9.3% (*Figure 3C*). The combined PVE by the additive effects of all significant markers for each taxon ranged from 0.000018 to 41.6%, with an average of 12% (*Figure 1A–B*). As expected, the combined additive effects of significant loci are typically much lower than the $h^2$, which is the upper bound as it represents the total additive genetic effect. Interestingly, there are several taxa for which the PVE by additive and dominance effects of all significant SNPs exceeds $h^2$ (e.g. genus *Odoribacter* and ASV234 [unclassified Ruminococcaceae] for RNA-based traits), indicating there are strong dominance effects. For example, *Odoribacter* has two significant regions which show overdominance, and ASV234 has seven significant regions which show underdominance.

## Functional annotation of candidate genes

Our mapping procedure involves testing for associations between a given microbial trait and a single SNP marker. However, the true genetic architecture is likely more complex. For example, multiple interacting genes in a common host regulatory pathway could influence a given taxonomic group and/or function, the latter of which could also be distributed among unrelated taxa (i.e. functional redundancy). Thus, in order to reveal potential higher-level biological phenomena among the identified loci, we performed pathway analysis to identify interactions and functional categories enriched among the genes in significant intervals. We used STRING (*Szklarczyk et al., 2019*) to calculate a protein-protein interaction (PPI) network of 925 protein-coding genes nearest to significant SNPs (upstream and/or downstream). A total of 768 genes were represented in the STRING database, and

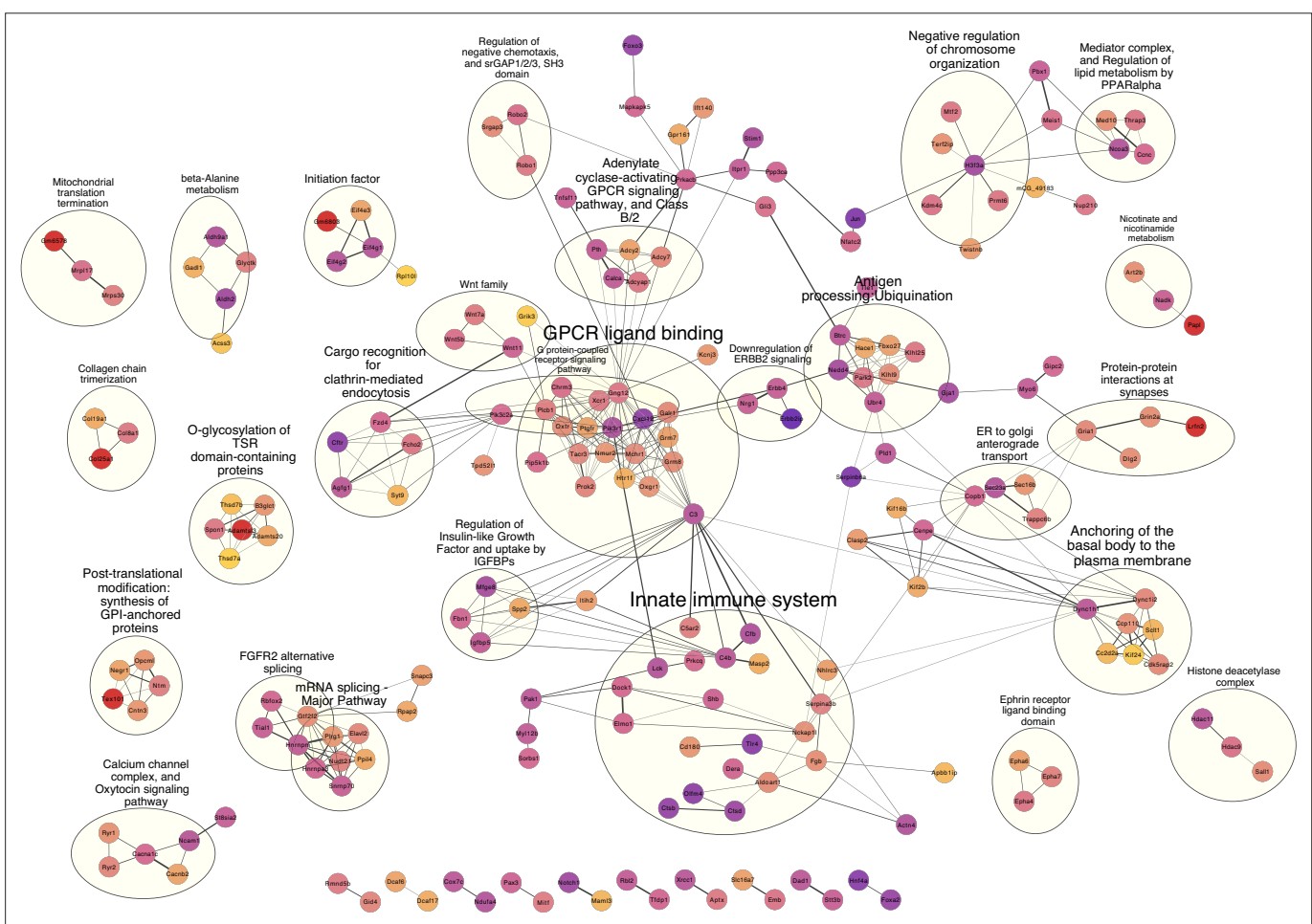

**Figure 4.** High confidence protein-protein interaction (PPI) network of genes closest to single nucleotide polymorphisms (SNPs) significantly associated with bacterial abundances. Network clusters are annotated using STRING's functional enrichment (*Doncheva et al., 2019*). Nodes represent proteins and edges their respective interactions. Only edges with an interaction score higher than 0.9 are retained. The width of the edge line expresses the interaction score calculated by STRING. The colour of the nodes describes the expression of the protein in the intestine where yellow is not expressed and purple is highly expressed.

The online version of this article includes the following figure supplement(s) for figure 4:

**Figure supplement 1.** Protein-protein interaction (PPI) network of hub genes of the 'nearest gene' network.

**Figure supplement 2.** Genes belonging to overrepresented KEGG pathways within the host genes closest to significant single nucleotide polymorphisms (SNPs) from association analysis.

**Figure supplement 3.** Enriched KEGG pathways.

**Figure supplement 4.** Enriched human diseases among genes closest to significant single nucleotide polymorphisms (SNPs) from association analysis.

the maximal network is highly significant (STRING PPI enrichment p-value: 2.15×10$^{-14}$) displaying 668 nodes connected by 1797 edges and an average node degree of 4.68. After retaining only the edges with the highest confidence (interaction score >0.9), this results in one large network with 233 nodes, 692 edges and ten smaller networks (*Figure 4*).

Next, we functionally annotated clusters using STRING's functional enrichment plugin. The genes of the largest cluster are part of the G-protein-coupled receptor (GPCR) ligand binding pathway. GPCRs are the largest receptor superfamily and also the largest class of drug targets (*Sriram and Insel, 2018*). We then calculated the top ten hub proteins from the network based on Maximal Clique Centrality (MCC) algorithm with CytoHubba to predict important nodes that can function as 'master switches' (*Figure 4—figure supplement 1*). The top ten proteins contributing to the PPI network were GNG12, MCHR1, NMUR2, PROK2, OXTR, XCR1, TACR3, CHRM3, PTGFR, and C3, which are all involved in the GPCR signalling pathway.

Furthermore, we performed enrichment analysis on the 925 genes nearest to significant SNPs using the clusterprofiler R package. We found 14 KEGG pathways to be overrepresented: circadian entrainment, oxytocin signalling pathway, axon guidance, calcium signalling, cAMP signalling, cortisol synthesis and secretion, cushing syndrome, gastric acid secretion, glutamatergic synapse, mucin type O-glycan biosynthesis, inflammatory mediator regulation of TRP channels, PD-L1 expression and the PD-1 checkpoint pathway in cancer, tight junction, and the *Wnt* signalling pathway (*Supplementary file 5*, *Figure 4—figure supplement 2* and *Figure 4—figure supplement 3*). Finally, genes involved in five human diseases are enriched, among them mental disorders (*Figure 4—figure supplement 4*).

Finally, due to the observation of a significant enrichment of cospeciating taxa among the bacterial species depleted in early onset IBD (*Groussin et al., 2017*) and the evidence that IBD is especially associated with a dysbiosis in mucosa-associated communities (*Yang et al., 2020a*; *Daniel et al., 2021*), we specifically examined possible overrepresentation of genes involved in IBD (*Khan et al., 2021*) among the 925 genes neighbouring significant SNPs. We found 14 out of the 289 IBD genes, which was significantly more than expected by chance (10,000 times permuted mean: 2.7, simulated p=00001, Fisher's exact test; *Supplementary file 6*). Interestingly, SNPs in 5 out of the 14 genes are associated with ASVs belonging to the genus *Oscillibacter,* a cospeciating taxon known to decrease during the active state of IBD (*Metwaly et al., 2020*).

## Comparison of significant loci to published gut microbiome mapping studies

Host loci that appear in multiple independent studies are more likely to represent true positive associations and/or less dependent on environmental perturbations. We therefore compiled a list of 648 unique confidence intervals of significant associations with gut bacterial taxa from seven previous mouse QTL studies (*Benson et al., 2010*; *McKnite et al., 2012*; *Leamy et al., 2014*; *Wang et al., 2015*; *Org et al., 2015*; *Snijders et al., 2016*; *Kemis et al., 2019*) and compared this list to our significance intervals for bacterial taxa at both the DNA and RNA level (341 intervals). Regions larger than 10 Mb were removed from all studies. We found 441 overlapping intervals, which is significantly more than expected by chance (10,000 times permuted mean: 372.8, simulated p=0.005, Fisher's exact test, see Materials and methods). Several of our smaller significant loci overlapped with larger loci from previous studies and removing this redundancy left 190 significant loci with a median interval size of 0.83 Mb (*Figure 5*). The most frequently identified locus is located on chromosome 2 169–171 Mb where protein coding genes *Gm11011, Znf217, Tshz2, Bcas1, Cyp24a1, Pfdn4, 4930470P17Rik*, and *Dok5* are situated.

Additionally, we collected genes within genome-wide significant regions reported in seven human microbiome GWAS (mGWAS) (*Bonder et al., 2016*; *Turpin et al., 2016*; *Goodrich et al., 2016*; *Wang et al., 2016*; *Hughes et al., 2020*; *Rühlemann et al., 2021*; *Kurilshikov et al., 2021*). However, no significant overrepresentation of genes was found within our significance intervals (p=*0*.16, Fisher's exact test*),* nor within our list of genes closest to a significant SNP (p=*0*.62, Fisher's exact test).

## Proteins differentially expressed in germ-free vs. conventional mice

To further validate our results, we compared the list of genes contained within intervals of our study to a list of differentially expressed proteins between germ-free and conventionally raised mice (*Mills et al., 2020*). This comparison was made based on the general expectation that if a host gene influences

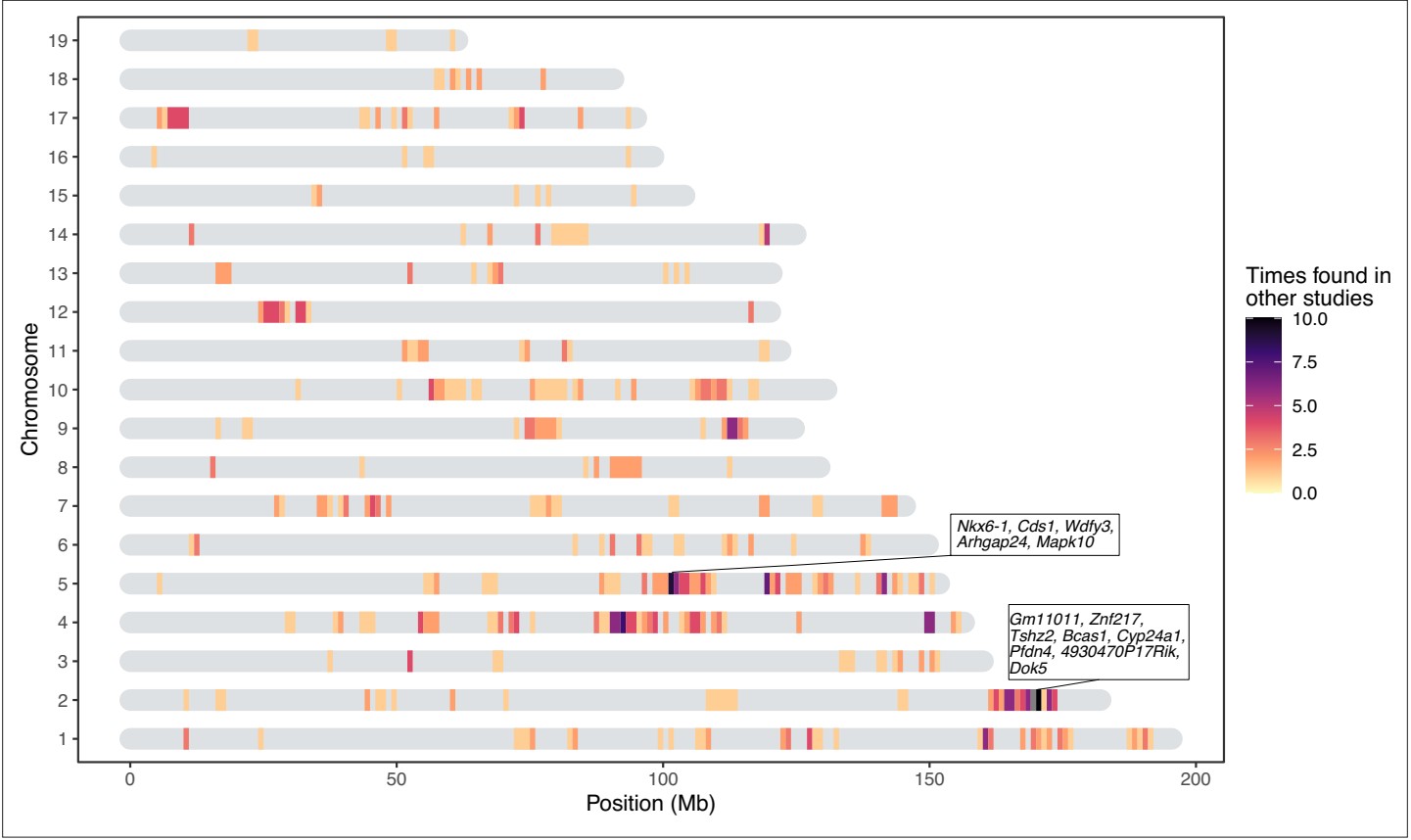

**Figure 5.** Significant loci in this study previously found in quantitative trait loci (QTL) studies of the mouse gut microbiome. The genes present in two repeatedly identified regions are depicted in boxes.

The online version of this article includes the following figure supplement(s) for figure 5:

**Figure supplement 1.** Protein-protein interaction (PPI) network of overlapping differentially expressed proteins according to colonization status.

**Figure supplement 2.** Protein-protein interaction network of hub genes of 'differentially expressed according to colonization status'-network.

microbial abundance, its own expression would be more likely to change according to differences in the microbiome. Thus, we examined the intersection between genes identified in our study and the proteins identified as highly associated ($|\pi|>1$) with the colonisation state of the colon and the small intestine (*Mills et al., 2020*). Out of the 373 overexpressed or underexpressed proteins according to colonisation status, we find 194 of their coding genes to be among our significant loci, of which 17 are the closest genes to a significant marker (*Iyd, Nln, Slc26a3, Slc3a1, Myom2, Nebl, Tent5a, Fxr1, Cbr3, Chrodc1, Nucb2, Arhgef10l, Sucla2, Enpep, Prkcq, Aacs,* and *Cox7c*). This is significantly more than expected by chance (simulated p=0.016, 10,000 permutations, Fisher's exact test). Furthermore, analysing the PPI with STRING results in a significant network (STRING PPI enrichment <i>P-value = $1.73 \times 10^{-14}$, and average node degree 2.4, *Figure 5—figure supplement 1*), with *Cyp2c65, Cyp2c55, Cyp2b10, Gpx2, Cth, Eif3k, Eif1, Sucla2,* and *Rpl17* identified as hub genes (*Figure 5—figure supplement 2*).

Subsequently, we merged the information from *Mills et al., 2020* and the seven previous QTL mapping studies discussed above to further narrow down the most promising candidate genes, and found 30 genes overlapping with our study. Of these 30 genes, six are the closest gene to a significant SNP. These genes are myomesine 2 (*Myom2*), solute carrier family 3 member 1 (*Slc3a1*), solute carrier family 26 member 3 (*Slc26a3*), nebulette (*Nebl*), carbonyl reductase 3 (*Cbr3*), and acetoacetyl-coA synthetase (*Aacs*).

## Candidate genes influencing bacterial abundance

To compile a comprehensive set of promising candidate genes, we combined results from network analysis, overlap with previous mouse QTL studies, and differential expression in GF vs conventional mice (see Materials and methods 'Curation of candidate genes' and ). Next, we used STRING to construct a PPI network with this curated gene set, which is highly significant (STRING PPI enrichment P-value < 1.0 × 10⁻¹⁶, average node degree = 4.85). We identified genes with the highest connectivity and most supporting information (original network see *Figure 6—figure supplement 1*), resulting in a final set of 79 candidate genes (*Figure 6* and *Supplementary file 7*). The G-protein, GNG12 and the complement component 3 C3, are the proteins with the most edges in the network (30 and 25, respectively), followed by MCHR1, CXCL12, and NMUR2 with each 18 edges. Of these 79 highly connected genes, 35 are associated with bacteria that are either cospeciating (cospeciation rate >0.5; *Groussin et al., 2017*) and/or have high heritability (>0.5) suggesting a functionally important role for these bacterial taxa.

## Discussion

Understanding the forces that shape variation in host-associated bacterial communities within host species is key to understanding the evolution and maintenance of meta-organisms. Although numerous studies in mice and humans demonstrate that host genetics influences gut microbiota composition (*McKnite et al., 2012*; *Leamy et al., 2014*; *Goodrich et al., 2014*; *Org et al., 2015*; *Davenport et al., 2015*; *Wang et al., 2016*; *Bonder et al., 2016*; *Goodrich et al., 2016*; *Kemis et al., 2019*; *Suzuki et al., 2019*; *Ishida et al., 2020*; *Hughes et al., 2020*; *Rühlemann et al., 2021*), our study is unique in a number of important ways. First, the unique genetic resource of mice collected from a naturally occurring hybrid zone together with their native microbes yielded extremely high mapping resolution and the possibility to uncover ongoing evolutionary processes in nature. Second, our study is the first to perform genetic mapping of 16S rRNA transcripts in the gut environment, which was previously shown to be superior to DNA-based profiling in a genetic mapping study of the skin microbiota (*Belheouane et al., 2017*). Third, our study is one of the only to specifically examine the mucosa-associated community. It was previously reasoned that the mucosal environment may better reflect host genetic variation (*Spor et al., 2011*), and evidence for this hypothesis exists in nature (*Linnenbrink et al., 2013*). Finally, by cross-referencing our results with previous mapping studies and recently available proteomic data from germ-free versus conventional mice, we curated a more reliable list of candidate genes and pathways. Taken together, these results provide unique and unprecedented insight into the genetic basis for host-microbe interactions (*Supplementary file 1*).

Importantly, by using wild-derived hybrid inbred strains to generate our mapping population, we gained insight into the evolutionary association between hosts and their microbiota at the transition from within species variation to between species divergence. Genetic relatedness in our mapping population significantly correlates with microbiome similarity, supporting a basis for codiversification at the early stages of speciation. A substantial proportion of microbial taxa are heritable, and heritability is correlated with cospeciation rates. This suggests that (i) vertical transmission could enable greater host adaptation to bacteria and/or (ii) the greater number of host genes associated with cospeciating taxa could indicate a greater dependency on the host, such that survival outside a specific host is reduced, making horizontal transmission less likely.

By performing 16S rRNA gene profiling at both the DNA and RNA level, we found that 14% (DNA based) to 20% (RNA based) of bacterial taxa have significant heritability estimates, with values up to 83%. The proportion of heritable taxa and maximum value of heritability estimates are consistent with previous studies in humans and mice ( *O'Connor et al., 2014*; *Org et al., 2015*; *Hughes et al., 2020*). Several factors of our study design likely contribute to high heritability values for some taxa. First, mice were raised in a controlled common environment, and heritability estimates in other mammals were shown to be contingent on the environment (*Grieneisen et al., 2021*). Second, bacterial communities were sampled from cecal tissue (mucosa) instead of lumen/faeces (*Linnenbrink et al., 2013*). Third, genetic variation in our mapping population was higher than in typical mapping studies due to subspecies differences. Finally, not all studies discriminate between narrow-sense heritability estimates and chip heritability estimates. Chip heritability represents only the variance explained by genotyped SNPs, thus estimates are lower than narrow-sense heritability estimates, which include

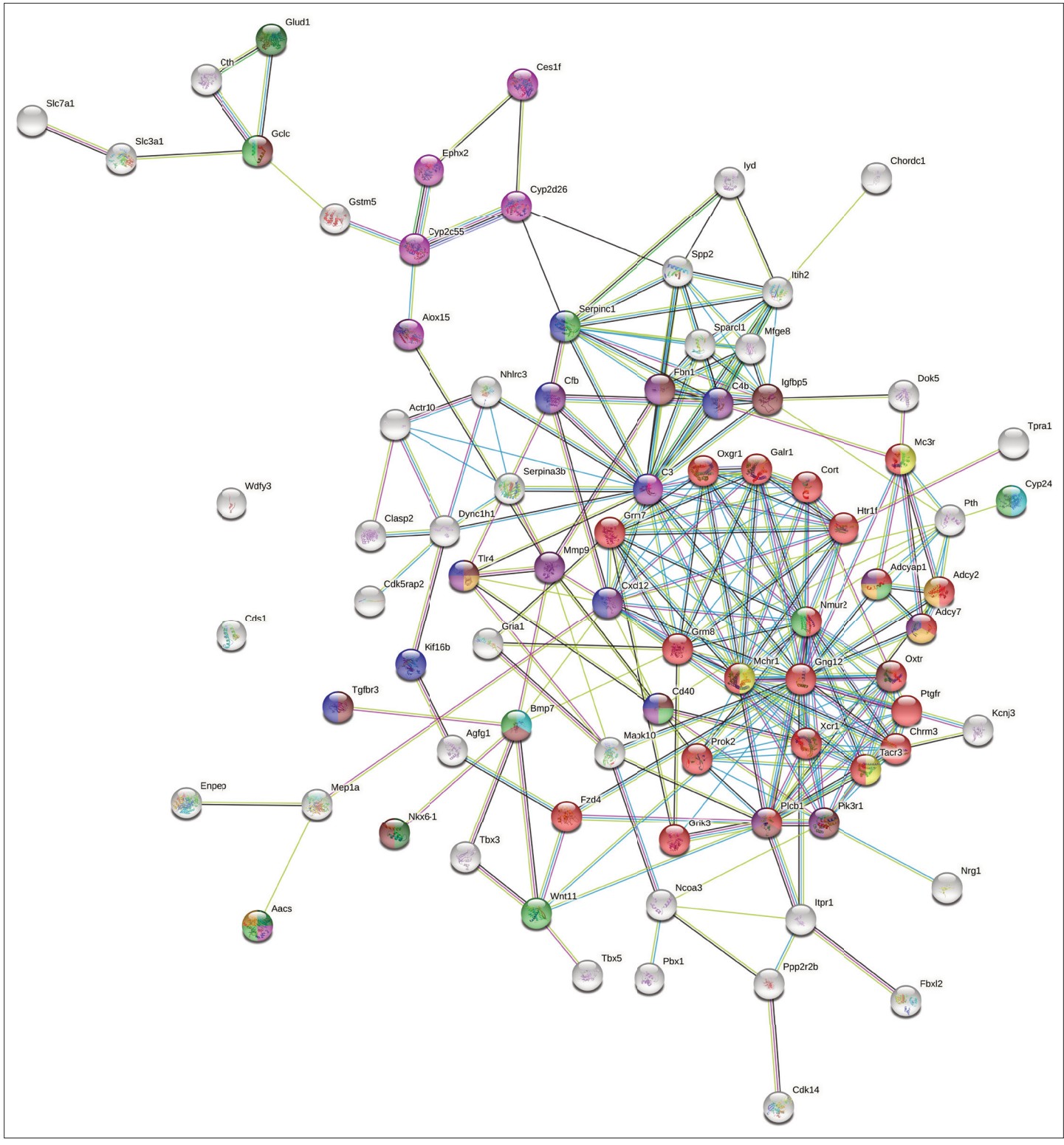

**Figure 6.** Network of host candidate genes influencing bacterial traits using STRING (https://string-db.org). The nodes represent proteins and are coloured according to a selection of enriched GO terms and pathways: G-protein coupled receptor (GPCR) signalling (red), regulation of the immune system process (blue), response to nutrient levels (light green), fatty acid metabolic process (pink), glucose homeostasis (purple), response to antibiotic (orange), regulation of feeding behaviour (yellow), positive regulation of insulin secretion (dark green), circadian entrainment (brown), and response to vitamin D (turquoise). The colour of the edges represents the interaction type: known interactions from curated databases (turquoise) or experimentally

*Figure 6 continued*

determined (pink); predicted interactions from gene neighbourhood (green), gene fusions (red), gene co-occurrence (blue); other interactions from text-mining (light green), coexpression (black), and protein homology (purple).

The online version of this article includes the following figure supplement(s) for figure 6:

**Figure supplement 1.** Protein protein interaction (PPI) network of 304 filtered candidate genes.

all additive genetic effects (*Zhou et al., 2013*). Chip heritability is particularly useful for phenotype prediction, for example, in the context of animal breeding and human disease, whereas narrow-sense heritability is useful for understanding genetic influence more broadly.

Notably, heritability estimates for RNA-based traits are significantly correlated with previously reported cospeciation rates in mammals (*Groussin et al., 2017*). This pattern, as well as the higher proportion of heritable taxa for RNA-based traits, suggest that host genetic effects are more strongly reflected by bacterial activity than cell number.

We found a total of 172 and 305 unique significant loci for DNA-based and RNA-based bacterial abundance, respectively, passing the conservative study-wide significance threshold. Taxa had a median of four significant loci, suggesting a complex and polygenic genetic architecture affecting bacterial abundances. We identify a higher number of loci in comparison to previous QTL and GWAS studies in mice (*Benson et al., 2010*; *McKnite et al., 2012*; *Leamy et al., 2014*; *Wang et al., 2015*; *Org et al., 2015*; *Snijders et al., 2016*; *Kemis et al., 2019*), which may be due to a number of factors. The parental strains of our study were never subjected to rederivation and subsequent reconstitution of their microbiota, and natural mouse gut microbiota are more variable than the artificial microbiota of laboratory strains (*Kohl and Dearing, 2014*; *Weldon et al., 2015*; *Suzuki, 2017*; *Rosshart et al., 2017*). Furthermore, as noted above, our mapping population harbours both within-subspecies and between-subspecies genetic variation. We crossed incipient species sharing a common ancestor ~0.5 million years ago, hence we may also capture the effects of mutations that fixed rapidly between subspecies due to strong selection, which are typically not variable within species (*Walsh and Lynch, 1998*; *Barton and Keightley, 2002*).

Importantly, our results also help to describe general features of the genetic architecture of bacterial taxon activity. For the majority of loci, the allele associated with lower relative abundance of the bacterial taxon was (partially) dominant. This suggests there is strong purifying selection against a high abundance of any particular taxon, which may help ensure high alpha diversity. For several bacterial taxa, the PVE by additive and dominance effects of significant SNPs exceeds the narrow-sense heritability, and the heterozygotes of one-third of significant SNPs displayed transgressive phenotypes. This is consistent with previous studies of hybrids (*Turner et al., 2012*; *Turner and Harr, 2014*; *Wang et al., 2015*), for example, wild-caught hybrids showed broadly transgressive gut microbiome phenotypes. This pattern can be explained by overdominance or underdominance, or by epistasis (*Rieseberg et al., 1999*). A curious observation is that *domesticus* alleles are associated with higher relative bacterial abundances twice as often as *musculus* alleles (*Figure 3A*). Although the biological explanation for this pattern is not discernible from our current results, future work incorporating quantitative profiling of bacterial load in hybrid and unadmixed *musculus* and *domesticus* individuals may help explain this phenomenon.

Notably, many loci significantly associated with bacterial abundance in this study were implicated in previous studies (*Figure 5*). For example, chromosome 2 169–171 Mb is associated with ASV23 (*Eisenbergiella*), *Eisenbergiella* and ASV32 (unclassified Lachnospiraceae) in this study, and overlaps with significant loci from three previous studies (*Leamy et al., 2014*; *Snijders et al., 2016*; *Kemis et al., 2019*). This region contains eight protein-coding genes: *Gm11011*, *Znf217*, *Tshz2*, *Bcas1*, *Cyp24a1*, *Pfdn4*, *4930470P17Rik*, and *Dok5*. Another hotspot is on chromosome 5 101–103 Mb. This locus is significantly associated with four taxa in this study (*Prevotellaceae*, *Paraprevotella*, ASV7 genus *Paraprevotella* and *Acetatifactor*) and overlaps with associations for Clostridiales, Clostridiaceae, Lachnospiraceae, and Deferribacteriaceae (*Snijders et al., 2016*). Protein-coding genes in this region are: *Nkx6-1*, *Cds1*, *Wdfy3*, *Arhgap24*, and *Mapk10*. As previous studies were based on rederived mouse strains, identifying significant overlap in the identification of host loci suggests that some of the same genes and/or mechanisms influencing major members of gut microbial communities are conserved even in the face of community 'reset' in the context of rederivation. The identity of the taxa is however not always the same, which suggests that functional redundancy may contribute to these

observations, if, for example, several bacterial taxa fulfill the same function within the gut microbiome (*Moya and Ferrer, 2016*; *Tian et al., 2020*).

A limitation of the current and previous genetic studies is that the phenotypes used for mapping are based on relative abundance rather than absolute, quantitative estimates. Specifically, an increase in a given taxon's abundance will necessarily lead to a decrease in abundance among all other taxa, which can lead to a number of potential biases (*Kathagen et al., 2017*; *Barlow et al., 2020*). Thus, future studies incorporating absolute abundance estimates may improve the detection of host-microbe interactions.

Nevertheless, our results do display overlap with studies based on other independent methods, such as a list of proteins differentially expressed in the intestine of germ-free mice compared to conventionally raised mice (*Mills et al., 2020*). Furthermore, by analysing the functions of the genes closest to significant SNPs, we found that 12 of the 14 significantly enriched KEGG pathways were shown to be related to interactions with bacteria (*Fonken et al., 2010 Thaiss et al., 2014*; *Neumann et al., 2014*; *Thaiss et al., 2015a*; *Thaiss et al., 2015b*; *Castoldi, 2015*; *Erdman and Poutahidis, 2016*; *Thaiss et al., 2016*; *Deaver et al., 2018*; *Wu et al., 2018*; *Peng et al., 2020*; *Nagpal et al., 2020*; *Hollander and Kaunitz, 2019*; *Supplementary file 5*).

To improve the robustness of our results, we combined multiple lines of evidence to prioritise candidates, resulting in a network of 79 genes (*Supplementary file 7*). At the centre of this network is a set of 22 proteins involved in G-protein coupled receptor signalling (*Figure 6*, red nodes). MCHR1, NMUR2, and TACR3 (*Figure 6*, yellow) are known to regulate feeding behaviour (*Saito et al., 1999*; *Cardoso et al., 2012*; *Smith et al., 2019*), and CHRM3 to control digestion (*Gautam et al., 2006*; *Tanahashi et al., 2009*). Gut microbes can produce GPCR agonists to elicit host cellular responses (*Cohen et al., 2017*; *Colosimo et al., 2019*; *Chen et al., 2019*; *Pandey et al., 2019*). Thus, GPCRs may be key modulators of communication between the gut microbiota and host. Another interesting group of genes are those responding to nutrient levels (*Bmp7*, *Cd40*, *Aacs*, *Gclc*, *Nmur2*, *Cyp24a1*, *Adcyap1*, *Serpinc1*, and *Wnt11*) (*Sethi and Vidal-Puig, 2008*; *Peier et al., 2009*; *Townsend et al., 2012*; *Yi and Bishop, 2015*; *Shi and Tu, 2015*; *Toderici et al., 2016*; *Yasuda et al., 2021*; *Gastelum et al., 2021*), as gut microbiota affect host nutrient uptake (*Chung et al., 2018*). In addition, CYP24A1, BMP7, and CD40 respond to vitamin D. Previous studies identified vitamin D/the vitamin D receptor to play a role in modulating the gut microbiota (*Wang et al., 2016*; *Malaguarnera, 2020*; *Yang et al., 2020b*; *Singh et al., 2020*), and CD40 is known to induce a vitamin D dependent antimicrobial response through IFN-γ activation (*Klug-Micu et al., 2013*).

Another important category of candidate genes are those involved in immunity. Our most significant SNP was situated downstream of the *Tlr4* gene and was associated with the genus *Dorea* and several *Dorea* species. *Dorea* is a known short chain fatty acid producer (*Taras, 2002*; *Reichardt et al., 2018*) and interacts with tight junction proteins *Claudin-2* and *Occludin* (*Alhasson et al., 2017*). *Tlr4* is a member of the Toll-like receptor family, and has been linked with obesity, inflammation, and changes in the gut microbiota (*Velloso et al., 2015*). These combined results reflect an important role for *Dorea* in fatty acid harvesting and intestinal barrier integrity, both of which could act systemically to activate TLR4 and to promote metabolic inflammation (*Cani et al., 2008*; *Delzenne et al., 2011*; *Nicholson et al., 2012*). Moreover, the SNP with the second lowest p-value was associated with the same taxa and situated 181 kb upstream of *Irak4*. IRAK4 is rapidly recruited after TLR4 activation to enable downstream activation of the NFκB immune pathway. *Irak4* has previously been associated with a change in bacterial abundance using inbred mice (*McKnite et al., 2012*; *Org et al., 2015*).

Finally, we identified notable links between candidate genes and five human diseases (mental disorders, blood pressure finding, systemic arterial pressure, substance-related disorders, and atrial septal deficits; *Figure 4—figure supplement 4*). The connection to mental disorders is intriguing as involvement of the gut microbiota is suspected (*Kelly et al., 2015*; *Foster et al., 2017*; *Cox and Weiner, 2018*; *Chen et al., 2019*; *Sarkar et al., 2020*; *Parker et al., 2020*; *Flux and Lowry, 2020*). Taken together with our finding of an enriched set of GPCRs, this highlights the importance of host-microbial interplay along the gut-brain axis. Moreover, we also identify a significant overrepresentation of IBD genes (*Khan et al., 2021*) among the 925 genes nearest to significant SNPs (*Supplementary file 6*). Interestingly, SNPs in 5 out of 14 genes are associated with ASVs belonging to the genus *Oscillibacter*, a highly cospeciating taxon known to decrease during the active state of IBD (*Metwaly et al., 2020*).

In summary, our study provides a number of novel insights into the importance of host genetic variation in shaping the gut microbiome, in particular for cospeciating bacterial taxa. These findings provide an exciting foundation for future studies of the precise mechanisms underlying host-gut microbiota interactions in the mammalian gut and should encourage future genetic mapping studies that extend analyses to the functional metagenomic sequence level.

# Materials and methods

## Key resources table

| Reagent type (species) or resource | Designation | Source or reference | Identifiers | Additional information |
|---|---|---|---|---|
| strain, strain background (*Mus musculus musculus-Mus musculus-domesticus*, males) | (HZ)A-(HZ)H | This paper | | Second generation wild-derived inter-crossed hybrid mouse line, *Mus musculus musculus-Mus musculus domesticus* originating from four breeding stocks captured in the wild hybrid zone around Freising, Germany. |
| biological sample (*Mus musculus*) | Cecum tissue (mucosa) | This paper | | Collected and stored in RNAlater overnight. After RNAlater removal tissue was stored in –20°C |
| biological sample (*Mus musculus*) | Ear clips | This paper | | For genotyping |
| commercial assay or kit | Allprep DNA/RNA 96-well | Qiagen (Hilden, Germany) | Cat. No.: 80,284 | DNA/RNA extraction |
| commercial assay or kit | Lysing matrix E | MP Biomedical (Eschwege, Germany) | SKU:116914050-CF | Lysing |
| commercial assay or kit | High-capacity cDNA Reverse Transcription Kit | Applied Biosystems (Darmstadt, Germany) | Cat. No.: 368,814 | cDNA transcription |
| sequence-based reagent | 27 F | https://doi.org/10.1016/j.ijmm.2016.03.004 | PCR primers | Forward primer V1-V2 hypervariable region |
| sequence-based reagent | 338 R | https://doi.org/10.1016/j.ijmm.2016.03.004 | PCR primers | Reverse primer V1-V2 hypervariable region |
| software, algorithm | DADA2 | https://doi.org/10.1038/nmeth.3869 | | 16S rRNA gene processing |
| software, algorithm | phyloseq | https://doi.org/10.1371/journal.pone.0061217 | | 16S rRNA gene analysis |
| commercial assay or kit | DNAeasy Blood and Tissue 96-well | Qiagen (Hilden, Germany) | Cat. No.: 69,504 | DNA extraction ear clips for genotyping |
| Other | GigaMUGA | Neogen, Lincoln, NE | | Illumina Infinium II array containing 141,090 SNP probes |
| software, algorithm | plink 1.9 | https://doi.org/10.1186/s13742-015-0047-8 | | Quality control genotypes |
| software, algorithm | GEMMA (v 0.98.1) | https://doi.org/10.1038/ng.2310 | | Genetic relatedness matrix |
| software, algorithm | lme4QTL | https://doi.org/10.1186/s12859-018-2057-x | | SNP-based heritability, GWAS |
| software, algorithm | exactLRT (RLRsim, v 3.1–6) | *Scheipl et al., 2008* | | Significance heritability estimates |
| software, algorithm | inv.logit (Gtools, v 3.9.2) | *Grieneisen et al., 2021* | | Inverse logistic transformation |
| software, algorithm | matSpDlite | *Nyholt, 2019*; https://doi.org/10.1038/sj.hdy.6800717 | | Study-wide significance threshold |
| software, algorithm | biomaRt (mm10) | *Durinck et al., 2009* | | Gene annotation |
| software, algorithm | r.squaredGLMM (MuMIn, v 1.37.17) | *Kamil, 2020* | | Percentage of variance explained |
| software, algorithm | locateVariants (VariansAnnotation, v 1.34.0) | https://doi.org/10.1093/bioinformatics/btu168 | | Nearest gene |
| software, algorithm | STRING (v 11) | https://doi.org/10.1093/nar/gky1131 | | Protein-protein interaction networks |
| software, algorithm | Cytoscape (v 3.8.2) | https://doi.org/10.1101/gr.1239303 | | Network analysis and visualisation |

*Continued on next page*

*Continued*

| Reagent type (species) or resource | Designation | Source or reference | Identifiers | Additional information |
|---|---|---|---|---|
| software, algorithm | MCODE | https://doi.org/10.1186/1471-2105-4-2 | | Cytoscape plugin for identifying network clusters |
| software, algorithm | stringApp | https://doi.org/10.1021/acs.jproteome.8b00702 | | Cytoscape plugin for functional annotation |
| software, algorithm | clusterprofiler (v 3.16.1) | https://doi.org/10.1089/omi.2011.0118 | | Enrichment analysis |
| software, algorithm | *poverlap* | *Pedersen and Brown, 2013* | | To determine significant overlap between studies |
| software, algorithm | R (v 3.5.3) | https://www.R-project.org/ | | |

## Intercross design

We generated a mapping population using partially inbred strains derived from mice captured in the *M. musculus–M. m. domesticus* hybrid zone around Freising, Germany, in 2008 (*Turner et al., 2012*). Originally, four breeding stocks were derived from 8 to 9 ancestors captured from one (FS, HA, TU) or two sampling sites (HO), and maintained with four breeding pairs per generation using the HAN-rotation out-breeding scheme (*Rapp, 1972*). Eight inbred lines (two per breeding stock) were generated by brother/sister mating of the 8th generation lab-bred mice. We set up the cross when lines were at the 5th–9th generation of brother-sister meeting, with inbreeding coefficients of >82%.

We used power calculations to estimate the optimal cross design and sample size needed. We first set up eight G1 crosses, each with one predominantly *domesticus* line (FS, HO – hybrid index <50%; see below) and one predominantly musculus line (HA, TU – hybrid index >50%); each line was represented as a dam in one cross and sire in another (*Figure 1—figure supplement 6*). One line, FS5, had a higher hybrid index than expected, suggesting there was a misidentification during breeding (see genotyping below). Next, we set up G2 crosses in eight combinations (sub-crosses), such that each G2 individual has one grandparent from each of the initial four breeding stocks. We included 40 males from each sub-cross in the mapping population resulting in a total of 320 mice. Mice were kept together with male littermates until moved to single cages 1 week prior to sacrifice.

This study was performed according to approved animal protocols and institutional guidelines of the Max Planck Institute. Mice were maintained and handled in accordance with FELASA guidelines and German animal welfare law (Tierschutzgesetz § 11, permit from Veterinäramt Kreis Plön: 1401–144/PLÖ–004697).

## Sample collection

Mice were kept in the same room and caged together with littermates after weaning before being separated into single cages 1 week prior to dissection (to minimize effects of social dominance on fertility traits for a related study). Mice were sacrificed at 91±5 days by $CO_2$ asphyxiation. We recorded body weight, body length and tail length, and collected ear tissue for genotyping. The caecum was removed and gently separated from its contents through bisection and immersion in RNAlater (Thermo Fisher Scientific, Schwerte, Germany). After overnight storage in RNAlater at 4°C, the RNAlater was removed and tissue stored at –20°C.

## DNA extraction and sequencing

We simultaneously extracted DNA and RNA from caecum tissue samples using Qiagen (Hilden, Germany) Allprep DNA/RNA 96-well kits. All samples were extracted together in the same extraction round and hence timepoint. We followed the manufacturer's protocol, with the addition of an initial bead beating step using Lysing matrix E tubes (MP Biomedical, Eschwege) to increase cell lysis. We used caecum tissue because host genetics has a greater influence on the microbiota at this mucosal site than on the lumen contents (*Linnenbrink et al., 2013*). We performed reverse transcription of RNA with high-capacity cDNA transcription kits from Applied Biosystems (Darmstadt, Germany). We amplified the V1–V2 hypervariable region of the 16S rRNA gene using barcoded primers (27F-338R) with fused MiSeq adapters and heterogeneity spacers following the description in *Rausch et al., 2016* and sequenced amplicons with 250bp paired-reads on the Illumina MiSeq platform. Individual sequencing libraries were prepared in parallel for all DNA and RNA samples, respectively, resulting

in one MiSeq library each. Accordingly, all individual 16S rRNA gene profiles within a given DNA- or RNA-based mapping analysis were generated by a single sequencing run. Thus, only direct comparisons between DNA- and RNA-based traits could be possibly confounded by sequencing run, and all analyses were performed independently for these two categories.

## 16S rRNA gene analysis

We assigned sequences to samples by exact matches of MID (multiplex identifier, 10 nt) sequences processed 16S rRNA sequences using the DADA2 pipeline, implemented in the DADA2 R package, version 1.16.0 (*Callahan et al., 2016*). Processing of the raw reads with DADA2 was performed separately for the DNA- and RNA-based libraries. Briefly, raw sequences were trimmed and quality filtered with the maximum two 'expected errors' allowed in a read, paired sequences were merged, ASVs were inferred, and chimeras removed. The two libraries were merged and another round of chimera removal was performed. We classified taxonomy using the Ribosomal Database Project (RDP) training set 16 (*Cole et al., 2014*). Classifications with low confidence at the genus level (<0.8) were grouped in the arbitrary taxon 'unclassified_group'. For all downstream analyses, we rarefied samples to 10,000 reads each. Due to the quality filtering, we have phenotyping data for 286 individuals on DNA level, and 320 G2 individuals on RNA level.

We used the phyloseq R package (version 1.32.0) to estimate alpha diversity using the Shannon index and Chao1 index, and beta diversity using Bray-Curtis distance (*McMurdie and Holmes, 2013*). We defined core microbiomes at the DNA- and RNA-level, including taxa present in >25% of the samples and with median abundance of non-zero values > 0.2% for ASV and genus; and >0.5% for family, order, class, and phylum.

## Genotyping

We extracted genomic DNA from ear samples using DNAeasy Blood and Tissue 96 well kits (Qiagen, Hilden, Germany), according to the manufacturer's protocol. We sent DNA samples from 26 G0 mice and 320 G2 mice to GeneSeek (Neogen, Lincoln, NE) for genotyping using the Giga Mouse Universal Genotyping Array (GigaMUGA; *Morgan et al., 2015*), an Illumina Infinium II array containing 141,090 SNP probes. We quality-filtered genotype data using plink 1.9 (*Chang et al., 2015*); we removed individuals with call rates < 90% and SNPs that were: not bi-allelic, missing in > 10% individuals, with minor allele frequency < 5%, or Hardy-Weinberg equilibrium exact test *P*-values < 1e-10. A total of 64,103 SNPs and all but one G2 individual were retained. Prior to mapping, we LD-filtered SNPs with $r^2$ > 0.9 using a window of 5 SNPs and a step size of 1 SNP. We retain 32,625 SNPs for mapping.

## Hybrid index calculation

For each G0 and G2 mouse, we estimated a hybrid index – defined as the percentage of *M. m. musculus* ancestry. We identified ancestry-informative SNP markers by comparing GigaMUGA data from ten individuals each from two wild-derived outbred stocks of *M. m. musculus* (Kazakhstan and Czech Republic) and two of *M. m. domesticus* (Germany and France) maintained at the Max Planck Institute for Evolutionary Biology (L.M. Turner and B. Payseur, unpublished data). We classified SNPs as ancestry informative if they had a minimum of 10 calls per subspecies, the major allele differed between *musculus* and *domesticus*, and the allele frequency difference between subspecies was >0.3. A total of 48,361 quality-filtered SNPs from the G0/G2 genotype data were informative, including 8775 SNPs with fixed differences between subspecies samples.

## Estimation of relatedness among individuals

We computed a centred and a standardised relatedness matrix using the 32,625 filtered SNPs with GEMMA (v 0.98.1; *Zhou and Stephens, 2012*). The centred relatedness matrix was calculated with the formula:

$$Centered\ GRM = \frac{1}{p} \sum_{i=1}^{p} \left( x_i - 1_n \bar{x}_i \right) \left( x_i - 1_n \bar{x}_i \right)^T$$

The standardised relatedness matrix was calculated with the formula:

$$\text{Standardized GRM} = \frac{1}{p} \sum_{i=1}^{p} \frac{1}{v_{x_i}} \left( x_i - 1_n \bar{x}_i \right) \left( x_i - 1_n \bar{x}_i \right)^T$$

where **X** denotes the n×p matrix of genotypes, $x_i$ as its $i$th column representing the genotypes of $i$th SNP, $\bar{x}_i$ as the sample mean and $1_n$ as a n×1 vector of 1's, and $v_{x_i}$ as the sample variance of $i$th SNP.

## Heritability of microbial abundances

We calculated heritabilities for bacterial abundances using linear mixed models implemented in the lme4qtl R package (version 0.2.2; *Ziyatdinov et al., 2018*). We included mating pair nested within the subcross identifier (*Figure 1—figure supplement 6*) as random effects to control for maternal effects and population structure, respectively. The narrow-sense heritability ($h^2$) is expressed as:

$$h^2 = \frac{\sigma_g^2}{\sigma_g^2 + \sigma_m^2 + \sigma_s^2 + \sigma_e^2}$$

where $\sigma_g^2$ is the genetic variance estimated by the centred GRM, $\sigma_m^2$ variance of the mating pair component, $\sigma_s^2$ the variance due to the subcross identifier, and $\sigma_e^2$ the variance due to residual environmental factors.

We also estimated CH using the method from *Zhou et al., 2013*, which estimates the variance explained by genotyped SNPs. CH is estimated using the standardised GRM.

We determined significance of the heritability estimates using exact restricted likelihood ratio tests, following Supplementary Note 3 in *Ziyatdinov et al., 2018*, using the exactRLRT() function of the R package RLRsim (version 3.1–6; *Scheipl et al., 2008*). Correlation with cospeciation rates was calculated for taxa shared between studies using the Spearman's correlation test.

## Genome-wide association mapping

Prior to mapping, we inverse logistic transformed bacterial abundances using the inv.logit function from the R package gtools (version 3.9.2; *Grieneisen et al., 2021*).

We performed association mapping in the R package lme4qtl (version 0.2.2; *Ziyatdinov et al., 2018*) with the following linear mixed model:

$$y_i = \mu + a_i X_{ij}^a + d_i X_{ij}^d + Wu + e$$

where $y_i$ is the phenotypic value of the $j$th individual; $\mu$ is the mean, $X_{ij}^a$ the additive and $X_{ij}^d$ the dominance genotypic index values coded as for individual $j$ at locus $i$. $a$ and $d$ indicate fixed additive and dominance effects, $W$ indicates random effects mating pair and centred kinship matrix, plus residual error $e$. For mapping, we used the centred GRM as the SNP effect size does not depend on its minor allele frequency. Moreover, the centred GRM typically provides better control for population structure in lower organisms and both GRMs perform equally in humans (*Zhou and Stephens, 2012*).

We estimated additive and dominance effects separately because we expected to observe underdominance and overdominance in our hybrid mapping population, as well as additive effects, and aimed to estimate their relative importance. To model the additive effect (i.e. 1/2 distance between homozygous means), genotypes at each locus, $i$, were assigned additive index values ($X^a \in 1, 0, -1$) for AA, AB, BB, respectively, with A indicating the major allele and B the minor allele. To model dominance effects (i.e. heterozygote mean – midpoint of homozygote means), genotypes were assigned dominance index values ($X^d \in 0, 1$) for homozygotes and heterozygotes, respectively.

We included mating pair as a random effect to account for maternal effects and cage effects, as male litter mates are kept together in a cage after weaning. We included kinship coefficient as a random effect in the model to account for population and family structure. To avoid proximal contamination, we used a leave-one-chromosome-out approach, that is, when testing each single-SNP association we used a relatedness matrix omitting markers from the same chromosome (*Parker et al., 2014*). Hence, for testing SNPs on each chromosome, we calculated a centred relatedness matrix using SNPs from all other chromosomes with GEMMA (v0.98.1; *Zhou and Stephens, 2012*). We calculated p-values for single-SNP associations by comparing the full model to a null model excluding fixed effects. Code for performing the mapping is available at https://github.com/sdoms/mapping_scripts (copy archived at swh:1:rev:d085e7782e9ac85e264fc6b70a5058a53fd7e9fe, *Doms, 2022*).

The chi-squared test statistics needed for the calculation of the genomic inflation factors were computed from the p-values assuming one degree of freedom. The genomic inflation factor was defined as the median of the observed chi-squared test statistic divided by the expected median of the corresponding chi-squared distribution, and was computed for each trait separately. For traits with a genomic inflation factor ($\lambda_{GC}$) above the proposed threshold of 1.05, we applied genomic control by dividing the chi-squared test statistic with $\lambda_{GC}$ (*Devlin and Roeder, 1999*).

We evaluated significance of SNP-trait associations using two thresholds; first, we used a genome-wide threshold for each trait, where we corrected for multiple testing across markers using the Bonferroni method (*Abdi, 2007*). Second, as bacteria interact with each other within the gut as members of a community, bacterial abundances are non-independent, so we calculated a study-wide threshold dividing the genome-wide threshold by the number of effective taxa included. We used matSpDlite (*Nyholt, 2019*; *Li and Ji, 2005*; *Qin et al., 2020*) to estimate the number of effective bacterial taxa based on eigenvalue variance.

To estimate the genomic interval represented by each significant LD-filtered SNP, we report significant regions defined by the most distant flanking SNPs in the full pre-LD-filtered genotype dataset showing $r^2$ >0.9 with each significant SNP. We combined significant regions less than 10 Mb apart into a single region. Genes situated in significant regions were retrieved using biomaRt (*Durinck et al., 2009*) and the mm10 mouse genome.

## Percentage of variance explained

We estimated the percentage of variance explained by the lead SNP using a linear mixed model with the additive and dominance genotypes of the lead SNP included as a fixed effect and mating pair and kinship matrix as random effects in lme4QTL (*Ziyatdinov et al., 2018*). We used the r.squaredGLMM function from the MuMIn R package (v 1.37.17; *Kamil, 2020*) to calculate the marginal $R\_GLMM^2$, which represents the variance explained by the fixed effects (i.e. the genotype effect). The total variance explained by all significant SNPs was calculated similarly with the exception of including all significant SNPs as fixed effects instead of only the lead SNP. This was performed using the additive genotypes only and both the additive and the dominance genotypes.

## Dominance analyses

We classified dominance for SNPs with significant associations on the basis of the *d/a* ratio (*Falconer, 1996*) where *d* is the dominance effect, *a* the additive effect. As the expected value under purely additive effects is 0. As our mapping population is a multiparental-line cross, and not all SNPs were ancestry-informative with respect to *musculus/domesticus*, the sign of *a* effects is defined by the major allele within our mapping population, which lacks clear biological interpretation. To provide more meaningful values, we report *d/|a|*, such that a value of 1=complete dominance of the allele associated with higher bacterial abundance, and a value of –1=complete dominance of the allele associated with lower bacterial abundance. Values above 1 or below –1 indicate over/underdominance. We classified effects of significant regions the following arbitrary *d/|a|* ranges to classify dominance of significant regions (*Burke et al., 2002*; *Miller et al., 2014*): underdominant < –1.25, high abundance allele recessive between –1.25 and –0.75, partially recessive between –0.75 and –0.25, additive between –0.25 and 0.25, partially dominant between 0.25 and 0.75, dominant 0.75 and 1.25, and overdominant > 1.25.

## Gene ontology and network analysis

The nearest genes upstream and downstream of the significant SNPs were identified using the locateVariants() function from the VariantAnnotation R package (version 1.34.0; *Obenchain et al., 2014*) using the default parameters. A maximum of two genes per locus were included (one upstream, and one downstream of a given SNP).

To investigate functions and interactions of candidate genes, we calculated a PPI network with STRING version 11 (*Szklarczyk et al., 2019*), on the basis of a list of the closest genes to all SNPs with significant trait associations. We included network edges with an interaction score >0.9, based on evidence from fusion, neighbourhood, co-occurrence, experimental, text-mining, database, and coexpression. We exported this network to Cytoscape v 3.8.2 (*Shannon et al., 2003*) for identification

of highly interconnected regions using the MCODE Cytoscape plugin (*Bader and Hogue, 2003*), and functional annotation of clusters using the stringApp Cytoscape plugin (*Doncheva et al., 2019*).

We identified over represented KEGG pathways and human diseases using the clusterprofiler R package (version 3.16.1; *Yu et al., 2012*). p-values were corrected for multiple testing using the Benjamini-Hochberg method. Pathways and diseases with an adjusted p-value < 0.05 were considered overrepresented.

## Calculating overlap with other studies and overrepresentation of IBD genes

To test for significant overlap with loci identified in previous mapping studies and for overrepresentation of IBD genes, we used the tool *poverlap* (*Pedersen and Brown, 2013*) to compare observed overlap to random expectations based on 10,000 permutations of significant regions. Regio`ping regions using the locateVariants() function from the VariantAnnotation R package (version 1.34.0; *Obenchain et al., 2014*).

## Candidate gene curation

From the total set of 11,618 genes situated within significant regions, we identified a set of high-confidence genes, which met one or more of the following criteria: (1) hub genes or nearest neighbours in the 'closest gene network' (*Figure 4*, *Figure 4—figure supplement 1*), (2) genes in regions overlapping with QTL from previous mouse studies (*Figure 5*), (3) genes differentially expressed between germ-free and conventional mice (*Figure 5—figure supplement 1* and *Figure 5—figure supplement 2*; *Mills et al., 2020*). After filtering, the resulting set of 309 genes were given as input into STRING (*Szklarczyk et al., 2019*) to construct a PPI network (304/309 genes were represented in the database; *Figure 6—figure supplement 1*). Finally, we selected one top candidate gene per significant region on the basis of network properties (degree and number of nodes) and/or fitting the most of the above-mentioned criteria. In case of a tie, the gene with the highest intestinal expression score (source: STRING) is chosen. Nodes without any edges were removed.

## Acknowledgements

We thank Diethard Tautz for generous support of mouse breeding and Camilo Medina and the MPI-Plön mouse team for performing mouse husbandry, and Katja Cloppenborg-Schmidt and Dr Sven Künzel for their excellent technical assistance. We thank Jason Wolf for constructive feedback on the statistical models and heritability estimation. We thank Mathieu Groussin for assistance with cospeciation rate data. Research funding for this project was provided by the Deutsche Forschungsgemeinschaft (DFG, German Research Foundation), Project-ID 261376515 – Collaborative Research Center 1182, 'Origin and Function of Metaorganisms', Project A2 (J.F.B. and A.F.), Cluster of Excellence 2167 'Precision Medicine in Chronic Inflammation (PMI)' (grant no. EXC2167 to A.F. and J.F.B.) and TU 500/2–1 to L.M.T, and by the Max Planck Society (to D Tautz).

## Additional information

### Funding

| Funder | Grant reference number | Author |
| --- | --- | --- |
| Deutsche Forschungsgemeinschaft | 261376515 | John F Baines |
| Deutsche Forschungsgemeinschaft | TU 500/2-1 | Leslie M Turner |
| Deutsche Forschungsgemeinschaft | 390884018 | John F Baines |

The funders had no role in study design, data collection and interpretation, or the decision to submit the work for publication.

## Author contributions
Shauni Doms, Data curation, Formal analysis, Investigation, Methodology, Resources, Software, Validation, Visualization, Writing – review and editing, Writing – original draft; Hanna Fokt, Cecilia J Chung, Resources; Malte Christoph Rühlemann, Formal analysis, Software, Validation; Axel Kuenstner, Formal analysis, Methodology, Validation; Saleh M Ibrahim, Conceptualization, Methodology; Andre Franke, Conceptualization, Funding acquisition, Supervision; Leslie M Turner, Conceptualization, Data curation, Formal analysis, Funding acquisition, Investigation, Methodology, Software, Supervision, Validation, Visualization, Writing – review and editing, Resources; John F Baines, Conceptualization, Funding acquisition, Methodology, Project administration, Supervision, Writing – review and editing, Investigation, Resources

## Author ORCIDs
Shauni Doms http://orcid.org/0000-0002-2129-4138
Malte Christoph Rühlemann http://orcid.org/0000-0002-0685-0052
Axel Kuenstner http://orcid.org/0000-0003-0692-2105
Leslie M Turner http://orcid.org/0000-0002-5105-3546
John F Baines http://orcid.org/0000-0002-8132-4909

## Ethics
This study was performed according to approved animal protocols and institutional guidelines of the Max Planck Institute. Mice were maintained and handled in accordance with FELASA guidelines and German animal welfare law (Tierschutzgesetz § 11, permit from Veterinä;ramt Kreis Plö;n: 1401-144/PLÖ;-004697).

## Decision letter and Author response
Decision letter https://doi.org/10.7554/eLife.75419.sa1
Author response https://doi.org/10.7554/eLife.75419.sa2

# Additional files

## Supplementary files
• Transparent reporting form

• Supplementary file 1. Narrow-sense and chip heritability estimates from the present study and previous mQTL studies combined with PVE by additive and additive and dominance effects of significant SNPs, and cospeciation rates reported by Groussin et al., (2017).

• Supplementary file 2. Genome-wide significant associations.

• Supplementary file 3. Study-wide significant associations.

• Supplementary file 4. Study-wide significant association with an interval size smaller than 1Mb.

• Supplementary file 5. Over-represented KEGG pathways within the genes neighbouring the peak SNP.

• Supplementary file 6. IBD genes found within the study-wide significant regions.

• Supplementary file 7. List of promising candidate genes.

## Data availability
DNA- and RNA-based 16S rRNA gene sequences are available under project accession number PRJNA759194. Code is available at https://github.com/sdoms/mapping_scripts, (copy archived at swh:1:rev:d085e7782e9ac85e264fc6b70a5058a53fd7e9fe).

The following datasets were generated:

| Author(s) | Year | Dataset title | Dataset URL | Database and Identifier |
|---|---|---|---|---|
| Doms S, Fokt H, Rühlemann MC, Chung CJ, Künstner A, Ibrahim S, Franke A, Turner LM, Baines JF | 2022 | DNA- and RNA-based 16S rRNA gene sequences | https://www.ncbi.nlm.nih.gov/sra/?term=PRJNA759194 | SRA, PRJNA759194 |
| Doms S, Fokt H, Rühlemann MC, Chung CJ, Künstner A, Ibrahim S, Franke A, Turner LM, Baines JF | 2022 | Code | https://github.com/sdoms/mapping_scripts | Github, d085e77 |

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
