## [Editor Report]

This paper uses inbred hybrid mouse lines to estimate the heritability of the mucosa-associated microbiome and map variants in the mouse genome that are associated with the composition of the microbiome. The findings are of broad interest to microbiome researchers and improve on knowledge in the field, as the mapping design facilitates the identification of narrow association intervals and points to a novel correlation between heritability and cospeciation rates. The manuscript provides useful information about the approach to heritability estimation, allowing the results to be more readily placed in context. Congratulations on this important contribution to the literature.

---

## [Decision Letter]

**Decision letter after peer review:**

Thank you for submitting your article "Key features of the genetic architecture and evolution of host-microbe interactions revealed by high-resolution genetic mapping of the mucosa-associated gut microbiome in hybrid mice" for consideration by *eLife*. Your article has been reviewed by 3 peer reviewers, and the evaluation has been overseen by a Reviewing Editor and Wendy Garrett as the Senior Editor. The reviewers have opted to remain anonymous.

Essential revisions:

1. Demonstrate the robustness of the heritability analyses, including consideration of potential technical and biological confounding variables and ideally using complementary analyses (e.g., distinct methods for estimating heritability).

2. Demonstrate the robustness of the phylosymbiosis (heritability-cospeciation rate correlation) to controlling for taxon abundance.

3. Control for population structure are well as relatedness in the trait mapping analysis.

*Reviewer #1 (Recommendations for the authors):*

1. Heritability estimates in the study are quite high, including traits that are more highly heritable than even canonical high heritability traits in humans, like height. This could be accurate, particularly in a lab-controlled environment where genetic variation is ramped up by the crossing design and hybrid origins of the founder, but I think requires closer examination. In particular, I was surprised that there does not appear to be any adjustment for covariates in the heritability analysis: in my experience, batch and technical effects are omnipresent in genomic studies, including microbiome analysis, and can affect heritability estimates in unpredictable ways. Although sex and age are controlled for in the study design, are there differences in housing, diet, sample processing, library prep, or sequencing quality/read yield/lane/batch that account for measurable variance and need to be taken into account?

2. Related, part of the benefit of estimating heritability in a mixed effects models framework is the ability to flexibly control for other covariates. I don't see a benefit in using Mantel tests to correlate genetic relatedness and microbiome structure overall-as heritability is much more directly supported by direct estimation-and suggest removing this part of the analysis. I realize that the genetic relatedness-microbiome structure analysis might provide a "holistic" picture, but that could also be done by analyzing the top PCs of the microbiome data in a formal h2 analysis.

3. For the individual trait mapping analysis, I have some concerns that the models might not adequately control for background population structure. It's a good step to control for genetic structure with the K matrix, but I would like to see that the results are robust to including the top PCs of the genotype data in the model as additional fixed effects. Where possible, you could also consider using LD score regression to check that there is no unexpected inflation of test statistics.

4. In Figure 3 and lines 251 – 258, the results report PVE levels that are impossible (since more than 100% of variance can't be explained). While this is noted in the text, and some potential explanations are provided (lines 256 – 258), the current presentation isn't very useful because it's unclear how much trait variance is really being explained. A better alternative might be to construct the equivalent of polygenic scores for each trait and ask how much variance in each associated phenotype these composite scores explain (this approach would provide one value per mapped trait, upper-bounded by the overall trait heritability).

5. Finally, I lost the thread a bit when reading the second half of the results. All of these sections focus on one type of enrichment/annotation analysis or another. However, the hypotheses or models being tested in several of the analyses were not clear to me. E.g., why should we expect microbiome-associated genes to encode proteins that are networked to one another? Is the relationship to the GF versus conventional comparison based on the idea that if genes affect the microbiome, then changes in the microbiome should also affect the proteins those genes encode (not obvious to me)? Ultimately, what does it mean to be a "promising" candidate gene (promising for what kind of application), and how is that defined by these enrichment analyses?

Rather than having readers try to fill in the rationale for these analyses, it would be helpful to clarify the motivation in the paper itself-or even shorten these sections to focus on those where there is the clearest underlying model or hypothesis. This might also avoid diluting the more interesting, less speculative findings presented earlier.

*Reviewer #2 (Recommendations for the authors):*

Overall, this was an excellent study and I have few comments for improvement. A couple of points:

The association between heritability of taxa and their co-speciation rates is interesting, but I wonder whether this could be explained by an association between abundance and statistical power to detect heritability/co-speciation. Have the authors considered modeling heritability as a function of co-speciation rate and relative abundance? Such an analysis would be important to determine whether heritability and co-speciation rate are associated independently of abundance.

Relative abundances of microbes were used for all analyses (as opposed to absolute abundance quantification). Because findings based on relative abundances can be difficult to interpret, more discussion on this limitation would be helpful.

My understanding is that heritability estimates were generated only for ASV relative abundances. However, previous heritability studies have examined taxonomic levels from ASV to phylum. It would be interesting to extend the analyses presented to higher taxonomic levels beyond ASVs (eg species to phylum).

*Reviewer #3 (Recommendations for the authors):*

The main point of concern for me are the heritability estimates. The paper reports heritability results that may be a bit surprising considering the current knowledge and literature: the heritability values are very high, with several values around 90% or higher. This is unexpected, and I am not sure how this is reconciled with the expectation that most variation in the microbiome is environmental rather than genetic. Although some potential reasons are given in the Discussion (mice raised in a controlled environment, using cecal content, etc), it still makes me a bit uneasy to see such high heritability estimates. One potential way to approach this is to try a different statistical approach for calculating heritability. Another would be to compare heritability estimates from this study with estimates from other studies – by now there are quite a few studies that report microbiome-wide heritability estimates from humans, mice, and other host species. It could be useful to correlate the heritability estimates in the current study with those from these studies. Lastly, it would be good to compare microbiome heritability estimates from this study to heritability’s of other complex traits in the same system – are there other known phenotypes that have heritability estimates that are this high?

[Editors' note: further revisions were suggested prior to acceptance, as described below.]

Thank you for resubmitting your work entitled "Key features of the genetic architecture and evolution of host-microbe interactions revealed by high-resolution genetic mapping of the mucosa-associated gut microbiome in hybrid mice" for further consideration by *eLife*. Your revised article has been evaluated by Wendy Garrett (Senior Editor) and a Reviewing Editor.

All reviewers appreciate the thorough revisions and re-analyses, which have resulted in a much improved manuscript. However, there are a few remaining issues that need to be addressed, as outlined below (the first item is the most essential):

1) Address the concern of Reviewer 3 below, which concern inflated heritability values in the main text. While the correlations in h2 estimates between methods are reassuringly high, Figure 8 in the response to reviewers also shows that the h2 reported in the paper (from lme4QTL) are typically several-fold (and sometimes an order of magnitude) higher than obtained from three alternative methods. This information would not be available to readers in the manuscript's current form. The response to reviewers indicates that lme4qtl was chosen because it had a high R2 with the PVE explained by the additive effects of significant SNPs (Figure 4 in the response to reviewers). However, both GEMMA and sommer produce similar R2 values (and in one case, a higher R2 value) but much lower heritability estimates. Do these lower estimates also correlate with co-speciation rates?

2) Consider the polishing revisions to the code repository suggested by Reviewer 3.

3) Consider integrating the useful explanation in the response to reviewers about experimental design controls for technical and batch effects in the main manuscript, as future readers may have similar questions.

*Reviewer #2 (Recommendations for the authors):*

The authors addressed all of my comments from the previous round of review. The inclusion of multiple alternative calculations of heritability have improved the manuscript substantially.

All 16S rRNA gene sequence data have been deposited to NCBI.

*Reviewer #3 (Recommendations for the authors):*

I thank the authors for a comprehensive revision that addressed many of my concerns. It was good to see that the heritability values are correlated across different methods, and that several other methods produced heritability values smaller than those generated by lme4QTL. This, together with the fact that heritability values for some bacteria are higher than those for traits like length and weight, supports my notion that lme4QTL heritability values are overestimated. I am not sure what the reason is -- it's hard to know without spending time digging into the data, and I might not have the statistics background to advise on this -- but I am not confident that these values are robust. I would encourage the authors to investigate these analyses very thoroughly, making sure that all potential confounders are accounted for, the models are reasonable, and there are no artifacts in the data. I would suggest the text includes the results of heritability analysis with other approaches (in addition to lme4QTL) more prominently: Figure 1 should report and visualize heritability values from all methods used, and visualize the correlations between them. The text should describe these results, the methods used, the heritability values reported and their correlation. There should be a clear discussion about the possible reasons for the high heritability values (and why they are lower using other methods). I would also suggest including the analysis comparing heritability values across studies in the text, and include a visualization of these correlations, rather than just reporting the p-value.

Regarding code availability, I want to thank the authors for enhancing the README file on the github repository, which now provides a nice description of the pipeline and analysis steps. However, I am not sure if this is sufficient for readers who want to reproduce the results: looking at the code itself, it seems like there are commands to load scripts that are not included in the repository (e.g. the snp_heritability_lme4qtl.R script loads the script function_for_gemma.r that I couldn't find anywhere), and these scripts might not be able to be run on other machines. I recommend amending the github repository and scripts so that anyone who wishes to do so is able to run the analysis and reproduce the results.

---

## [Author Response]

Essential revisions:1. Demonstrate the robustness of the heritability analyses, including consideration of potential technical and biological confounding variables and ideally using complementary analyses (e.g., distinct methods for estimating heritability).2. Demonstrate the robustness of the phylosymbiosis (heritability-cospeciation rate correlation) to controlling for taxon abundance.3. Control for population structure are well as relatedness in the trait mapping analysis.Reviewer #1 (Recommendations for the authors):1. Heritability estimates in the study are quite high, including traits that are more highly heritable than even canonical high heritability traits in humans, like height. This could be accurate, particularly in a lab-controlled environment where genetic variation is ramped up by the crossing design and hybrid origins of the founder, but I think requires closer examination. In particular, I was surprised that there does not appear to be any adjustment for covariates in the heritability analysis: in my experience, batch and technical effects are omnipresent in genomic studies, including microbiome analysis, and can affect heritability estimates in unpredictable ways. Although sex and age are controlled for in the study design, are there differences in housing, diet, sample processing, library prep, or sequencing quality/read yield/lane/batch that account for measurable variance and need to be taken into account?

We thank the reviewer for raising this important concern and agree that the heritability estimates appear high. We accordingly performed several analyses, which do confirm the robustness of the results. Details are given below in response to comment 3 and the major issue of Reviewer 3. In short, this resulted in the inclusion of a “sub-cross identifier” as an additional random effect (see Figure 1—figure supplement 4 for cross design). The current model includes the sub-cross identifier with the mating pair identifier nested within it (Author response image 1). We now clarify this in the text (see lines 834-836). The mating pair id accounts for both maternal effects and cage effects, as the mice were kept together with littermates after weaning and separated into single cages one week prior to dissection. All mice were kept in the same room.

**Author response image 1. sa2fig1:** Visualization of the random effects: mating pair identifier (A) and mating pair identifier nested within the sub-cross ID (B).

We also agree that batch and technical effects are important to account for. Due to the setup of our extraction and sequencing library prep procedures, these factors were inherently accounted for at numerous levels. The samples were extracted using the Qiagen DNA/RNA Allprep kit in 96-well plate format, such that all samples were extracted in the same extraction round and hence timepoint. Individual sequencing libraries were subsequently prepared in parallel for all DNA and RNA samples, respectively, resulting in one MiSeq library each. Accordingly, all individual 16S rRNA gene profiles within a given DNA- or RNA-based mapping analysis were generated by a single sequencing run. Thus, only direct comparisons between DNA- and RNA-based traits could be possibly confounded by sequencing run, and all analyses were performed independently for these two categories.

2. Related, part of the benefit of estimating heritability in a mixed effects models framework is the ability to flexibly control for other covariates. I don't see a benefit in using Mantel tests to correlate genetic relatedness and microbiome structure overall-as heritability is much more directly supported by direct estimation-and suggest removing this part of the analysis. I realize that the genetic relatedness-microbiome structure analysis might provide a "holistic" picture, but that could also be done by analyzing the top PCs of the microbiome data in a formal h2 analysis.

Thank you for this suggestion, we agree that this result is not among the most important of our study and followed this advice as an opportunity to make the Results section more concise, also in reference to comment #5 of the same reviewer below.

3. For the individual trait mapping analysis, I have some concerns that the models might not adequately control for background population structure. It's a good step to control for genetic structure with the K matrix, but I would like to see that the results are robust to including the top PCs of the genotype data in the model as additional fixed effects. Where possible, you could also consider using LD score regression to check that there is no unexpected inflation of test statistics.

Thank you, we agree these suggestions would help show the robustness of the mapping analysis. Accordingly, we compared three different models and ran them on a representative subset of microbial traits (n=27 RNA-based genus level traits): (1) Current model with the mating pair and the kinship matrix as random effects, (2) Current model + PC1-PC3, (3) Current model + the mating pair id nested within the sub-cross identifier (see Figure 1—figure supplement 4 for cross design). We first compared the BIC scores of these models without a SNP effect (Author response table 1). We see that the BIC is, except for one trait (G_Odoribacter), the lowest (bold) in our current model.

**Author response table 1. sa2table1:** BIC scores models for RNA-based genera.

Taxa	Current	+PC1	+PC2	+PC3	+PC4	+PC5	Current +subcross
G_Acetatifactor	**-2651.9**	-2637.7	-2628.1	-2613.8	-2599.6	-2584.5	-2646.2
G_Alistipes	**-2714**	-2711.2	-2701.5	-2687.3	-2673.9	-2658.9	-2712.8
G_Anaerostipes	**-3416.6**	-3400.9	-3385.2	-3369.2	-3352.7	-3336.1	-3410.8
G_Bacteroides	**-2076.6**	-2065.9	-2053.9	-2045.2	-2033.5	-2021.5	-2070.9
G_Butyricicoccus	**-3676.1**	-3661.1	-3645.3	-3628.4	-3612.7	-3596.6	-3671.4
G_Clostridium_XlVa	**-2356.8**	-2354.2	-2340.6	-2327.2	-2314.8	-2301	-2351.1
G_Dorea	**-2979.7**	-2965.4	-2952	-2936.6	-2926.1	-2916.2	-2974.4
G_Eisenbergiella	**-1500.5**	-1492	-1489.3	-1479.8	-1469.3	-1458.2	-1495.3
G_Fusicatenibacter	**-2932.8**	-2918.3	-2902.8	-2888.3	-2873.5	-2857.8	-2927.1
G_Helicobacter	**-1145.4**	-1138.4	-1131	-1122.1	-1114.8	-1104.8	-1140.3
G_Hungatella	**-2415.2**	-2402.3	-2390.2	-2376.7	-2364.5	-2351.1	-2409.5
G_Intestinimonas	**-3286.5**	-3272.5	-3256.3	-3240.1	-3228.2	-3213.9	-3282.4
G_*Lactobacillus*	**-2407**	-2395.6	-2384.4	-2380.5	-2366.8	-2352.6	-2403.5
G_Marvinbryantia	**-2961.6**	-2950.7	-2934.7	-2918.8	-2903.2	-2887.3	-2956.4
G_Mucispirillum	**-1938.8**	-1930.8	-1918.9	-1907.3	-1896.7	-1884.8	-1933.1
G_Odoribacter	-3183.2	-3170.8	-3156	-3143.5	-3134.7	-3120	**-3186.1**
G_Oscillibacter	**-2117.2**	-2105.8	-2095.2	-2084.3	-2072.1	-2060	-2111.9
G_Paraprevotella	**-2230.2**	-2222.8	-2219.2	-2207.7	-2195.4	-2182.8	-2228.4
G_Roseburia	**-2719**	-2704	-2689.4	-2674.8	-2661.3	-2647.6	-2713.5
G_Staphylococcus	**-7666.6**	-7636.9	-7607.6	-7576.8	-7546.5	-7515.8	-7660.9
unclassified_C_Deltaproteobacteria	**-3208.6**	-3197.8	-3183.4	-3167.3	-3158.7	-3142.1	-3205.5
unclassified_F_Lachnospiraceae	**-1435.4**	-1433.5	-1423	-1413.9	-1407.2	-1396.8	-1433.1
unclassified_F_Porphyromonadaceae	**-2997.7**	-2984.6	-2970.3	-2956.3	-2943.4	-2928.9	-2991.9
unclassified_F_Prevotellaceae	**-4216.4**	-4198.5	-4182	-4175.1	-4156.4	-4136.9	-4212.7
unclassified_F_Ruminococcaceae	**-2590.6**	-2578.8	-2565.6	-2554.4	-2540.7	-2525.9	-2586.1
unclassified_O_Bacteroidales	**-2348**	-2339.5	-2328	-2317	-2305.3	-2291.7	-2344.4
unclassified_O_Clostridiales	**-2721.2**	-2708	-2694.8	-2681	-2667.4	-2654.7	-2715.4
unclassified_P_Bacteroidetes	**-3260.7**	-3252.8	-3237.9	-3222.6	-3207	-3194.2	-3256.9

Next, we calculated the genomic inflation factors (Author response image 2) as well as the LD score regression intercept (Author response image 3) for this subset of traits. Running an LD score regression on data sets with less than 200k SNPs is however not advised (Bulik-Sullivan et al., 2015). Both the genomic inflation factor and the LD score regression intercept show the lowest inflation in the current model. Some traits did however show some inflation of the test statistics above the proposed threshold of 1.05. For this reason, we used genomic control for the traits with a genomic inflation factor above 1.05 (lines 887-895). Following this correction, the total number of study-wide significant loci changed from 443 to 428; all downstream analyses were accordingly repeated with the final updated list of loci.

**Author response image 2. sa2fig2:** Genomic inflation factor for different P values (additive effect, left; dominance effect, middle; total model, right) calculated by the different models: current model including the mating pair id nested in the sub-cross ID (top), current model (middle), and the current model including PC1-PC3 as fixed effects (bottom). Dashed line and values indicate the average genomic inflation factor.

**Author response image 3. sa2fig3:** LD score regression intercept for P values calculated by the different models: current model including the mating pair id nested in the sub-cross ID (top), current model (middle), and the current model including PC1-PC3 as fixed effects (bottom). Dashed line and values indicate the average genomic inflation factor.

4. In Figure 3 and lines 251 – 258, the results report PVE levels that are impossible (since more than 100% of variance can’t be explained). While this is noted in the text, and some potential explanations are provided (lines 256 – 258), the current presentation isn’t very useful because it’s unclear how much trait variance is really being explained. A better alternative might be to construct the equivalent of polygenic scores for each trait and ask how much variance in each associated phenotype these composite scores explain (this approach would provide one value per mapped trait, upper-bounded by the overall trait heritability).

We thank the reviewer for this suggestion. We used a linear mixed model that includes all the significant SNPs per trait to calculate a polygenic score equivalent. We adapted the manuscript Figure 3 accordingly and added a section to the Methods (line 908-917):

"The total variance explained by all significant SNPs was calculated similarly with the exception of including all significant SNPs as fixed effects instead of only the lead SNP. This was performed using the additive genotypes only and both the additive and the dominance genotypes."

We used these "polygenic scores" to determine the choice of linear models and software/methods for calculating the heritability estimates. As seen in Author response image 4, using lme4QTL including the mating pair ID nested in the sub-cross ID (lme4qtl.fam.heritability) to estimate the heritability has a slope of 0.921, which is the closest to 1 of all the methods tested, and yields the highest *R^2^* value (0.14).

**Author response image 4. sa2fig4:** Correlation of heritability estimates calculated using different methods/programs and models to the percentage of phenotypic variance explained by the additive genotypes of all significant SNPs within the trait. For Gaston, Sommer, Gemma 1PC, and lme4QTL.1PC, a linear mixed model was used with the first PC of the genotypes as a fixed effect and a genomic relatedness matrix (GRM) as a random effect. For Gemma.3PC, the first three PCs of the genotype data were used as fixed effects and a GRM as a random effect. For lme4QTL.MP, the mating pair ID was used as a random effect together with the GRM, while in lme4QTL.fam the mating pair ID was nested in the sub-cross ID as random effects together with a GRM to estimate the additive heritability.

5. Finally, I lost the thread a bit when reading the second half of the results. All of these sections focus on one type of enrichment/annotation analysis or another. However, the hypotheses or models being tested in several of the analyses were not clear to me. E.g., why should we expect microbiome-associated genes to encode proteins that are networked to one another? Is the relationship to the GF versus conventional comparison based on the idea that if genes affect the microbiome, then changes in the microbiome should also affect the proteins those genes encode (not obvious to me)? Ultimately, what does it mean to be a "promising" candidate gene (promising for what kind of application), and how is that defined by these enrichment analyses?Rather than having readers try to fill in the rationale for these analyses, it would be helpful to clarify the motivation in the paper itself-or even shorten these sections to focus on those where there is the clearest underlying model or hypothesis. This might also avoid diluting the more interesting, less speculative findings presented earlier.

We thank the reviewer for this constructive input. We now added and/or re-worded text at the beginning of each respective subsection to more explicitly state the motivation/hypothesis behind these additional analyses, which we believe are ultimately important to draw higher-level conclusions/generate novel hypotheses to serve as a basis of future work (lines 379-383, lines 444-445, lines 476-480, lines 510-514).

Reviewer #2 (Recommendations for the authors):Overall, this was an excellent study and I have few comments for improvement. A couple of minor points:The association between heritability of taxa and their co-speciation rates is interesting, but I wonder whether this could be explained by an association between abundance and statistical power to detect heritability/co-speciation. Have the authors considered modeling heritability as a function of co-speciation rate and relative abundance? Such an analysis would be important to determine whether heritability and co-speciation rate are associated independently of abundance.

Thank you for this suggestion. We used a multiple linear regression model to test if the cospeciation rate and the median relative abundance significantly predict the heritability estimate. The overall regression model was not significant (*R^2^*=0.27, F_(2,17)_=3.202, *P*=.067). Further, the cospeciation rate significantly predicts the heritability estimate (*P*=.022), while the median abundance does not (*P*=.92). We thus conclude that the heritability estimates and the cospeciation rates are associated independent of the abundance of a taxon in question (lines 209-215).

Relative abundances of microbes were used for all analyses (as opposed to absolute abundance quantification). Because findings based on relative abundances can be difficult to interpret, more discussion on this limitation would be helpful.

We thank the reviewer for pointing out this important aspect. We added a specific passage in the Discussion section (lines 670-675):

“A limitation of the current and previous genetic studies is that the phenotypes used for mapping are based on relative abundance rather than absolute, quantitative estimates. Specifically, an increase in a given taxon’s abundance will necessarily lead to a decrease in abundance among all other taxa, which can lead to a number of potential biases (Vandeputte et al., 2017; Barlow et al., 2020). Thus, future studies incorporating absolute abundance estimates may improve the detection of host-microbe interactions.”

My understanding is that heritability estimates were generated only for ASV relative abundances. However, previous heritability studies have examined taxonomic levels from ASV to phylum. It would be interesting to extend the analyses presented to higher taxonomic levels beyond ASVs (eg species to phylum).

We calculated heritability estimates with the updated model for all taxonomic levels (see reply to the comments of Reviewers 1 and 3). The labels on Figure 1A-B of the article are colored by taxonomic level.

Reviewer #3 (Recommendations for the authors):The main point of concern for me are the heritability estimates. The paper reports heritability results that may be a bit surprising considering the current knowledge and literature: the heritability values are very high, with several values around 90% or higher. This is unexpected, and I am not sure how this is reconciled with the expectation that most variation in the microbiome is environmental rather than genetic. Although some potential reasons are given in the Discussion (mice raised in a controlled environment, using cecal content, etc), it still makes me a bit uneasy to see such high heritability estimates. One potential way to approach this is to try a different statistical approach for calculating heritability. Another would be to compare heritability estimates from this study with estimates from other studies – by now there are quite a few studies that report microbiome-wide heritability estimates from humans, mice, and other host species. It could be useful to correlate the heritability estimates in the current study with those from these studies. Lastly, it would be good to compare microbiome heritability estimates from this study to heritability’s of other complex traits in the same system – are there other known phenotypes that have heritability estimates that are this high?

We agree with the Reviewer that it is important to critically evaluate the robustness of heritabilities. Accordingly, we calculated the heritability estimates using gaston (Hervé and Claire, 2020), sommer (Covarrubias-Pazaran, 2018), and GEMMA (Zhou and Stephens, 2012), as well as lme4QTL (Ziyatdinov et al., 2018). To compare the heritability estimates between the different methods, we used the same model where the first genotype PC is included as a fixed effect and the genetic relatedness matrix (GRM) is included as a random effect, as GEMMA does not allow categorical covariates such as mating pair ID (Figure 9). We used identical GRMs in lme4QTL and GEMMA. However, sommer and gaston incorporate estimation of GRMs when calculating heritabilities.

Heritability estimates from each of the alternative methods are significantly correlated with estimates from lme4QTL (gaston R=0.52; GEMMA R=0.93; sommer R=0.79; *P* < 2.2 e-16, Spearman's correlation; Author response image 5). However, the heritability estimates calculated by lme4QTL are consistently higher than estimates from other methods. In the revised manuscript, we use lme4QTL to calculate the heritability’s with the mating pair nested in the sub-cross ID as random effects to control for maternal and population structure (see reply to comment 4 from Reviewer 1; Author response image 4).

**Author response image 5. sa2fig5:** Heritability estimates calculated with lme4QTL compared to other methods. The dashed line represents the identity line with a slope of 1. The colored lines are the corresponding linear regression lines.

This results in fewer taxa with significant heritability’s, and estimated values are lower. Importantly, the correlation of the cospeciation rate with the heritability estimates remains significant (*P* = .037, R=0.47)

Next, we compared our heritability estimates with other studies. Other microbiome QTL/GWAS studies in humans and mice also reported high heritability estimates for some taxa: 82 % in (O’Connor et al., 2014), 92% in (Org et al., 2015), and 99% in (Hughes et al., 2020). We calculated the correlation of our heritability estimates with seven human studies (Davenport et al., 2015; Goodrich et al., 2016; Turpin et al., 2016; Xu et al., 2020; Ishida et al., 2020; Hughes et al., 2020; Kurilshikov et al., 2021), two mouse studies (O’Connor et al., 2014; Org et al., 2015), one pig study (Chen et al., 2018), and one primate study (Grieneisen et al., 2021). We found our DNA-based heritability estimates to be positively correlated with DNA-based heritability estimates of male mice in (Org et al., 2015) (R=0.6047, *P*=.04872, n=11) and with the heritability estimates from one human study by (Turpin et al., 2016) (R=0.3760, *P*=.04867, n=28).

Finally, we estimated the heritability of other traits in our mapping population; estimates for body weight (87%) and body length (67%) are also high and comparable to previous studies in mice (Keenan et al., 2021). This has been added to results (lines 158-159, 193-200).

[Editors' note: further revisions were suggested prior to acceptance, as described below.]

All reviewers appreciate the thorough revisions and re-analyses, which have resulted in a much improved manuscript. However, there are a few remaining issues that need to be addressed, as outlined below (the first item is the most essential):1) Address the concern of Reviewer 3 below, which concern inflated heritability values in the main text. While the correlations in h2 estimates between methods are reassuringly high, Figure 8 in the response to reviewers also shows that the h2 reported in the paper (from lme4QTL) are typically several-fold (and sometimes an order of magnitude) higher than obtained from three alternative methods. This information would not be available to readers in the manuscript's current form. The response to reviewers indicates that lme4qtl was chosen because it had a high R2 with the PVE explained by the additive effects of significant SNPs (Figure 4 in the response to reviewers). However, both GEMMA and sommer produce similar R2 values (and in one case, a higher R2 value) but much lower heritability estimates. Do these lower estimates also correlate with co-speciation rates?

Thank you for the further input to improve our heritability results. After careful inspection, we determined that the analyses described in the previous reply (performed in response to original comments from Reviewer 1) were not appropriate. Including principle components of the SNP-based relatedness matrix as fixed effects in the model only makes sense if the kinship matrix included as a random effect is pedigree-based. This is not necessary because we include the full SNP-based relatedness matrix as a random effect in the model following the approach in GEMMA, because “using a lower-rank relatedness matrix compromises the ability of the linear mixed model to control for sample structure.” (Zhou and Stephens, 2012, Supplementary text).

We recalculated heritability estimates in GEMMA without including PCs, and address differences in heritability estimates in detail below, in response to Reviewer 3.

The previous RNA-based heritability estimates from GEMMA and sommer do significantly correlate with co-speciation rates (Author response images 6-8). In the revised manuscript, we report that both chip heritability (newly calculated as described below) and narrow-sense heritability estimates for RNA are correlated with co-speciation rates (lines 142 – 161 and 177-182).

**Author response image 6. sa2fig6:** Correlation of heritability estimates, calculated with GEMMA using the first principal component as a fixed effect, with co-speciation rates reported in Groussin et al., 2017.

**Author response image 7. sa2fig7:** Correlation of heritability estimates, calculated with GEMMA using the first three principal component as a fixed effect, with co-speciation rates reported in Groussin et al., 2017.

**Author response image 8. sa2fig8:** Correlation of heritability estimates, calculated with sommer using the first principal component as a fixed effect, with co-speciation rates reported in Groussin et al., 2017.

2) Consider the polishing revisions to the code repository suggested by Reviewer 3.

Please see reply below.

3) Consider integrating the useful explanation in the response to reviewers about experimental design controls for technical and batch effects in the main manuscript, as future readers may have similar questions.

Thank you for this advice. We added the explanation in the respective Methods sections (Sample collection, lines 614-616; DNA extraction and sequencing, lines 624-625 and 634-639; 16S rRNA gene analysis lines 643-655).

Reviewer #3 (Recommendations for the authors):I thank the authors for a comprehensive revision that addressed many of my concerns. It was good to see that the heritability values are correlated across different methods, and that several other methods produced heritability values smaller than those generated by lme4QTL. This, together with the fact that heritability values for some bacteria are higher than those for traits like length and weight, supports my notion that lme4QTL heritability values are overestimated. I am not sure what the reason is -- it's hard to know without spending time digging into the data, and I might not have the statistics background to advise on this -- but I am not confident that these values are robust. I would encourage the authors to investigate these analyses very thoroughly, making sure that all potential confounders are accounted for, the models are reasonable, and there are no artifacts in the data. I would suggest the text includes the results of heritability analysis with other approaches (in addition to lme4QTL) more prominently: Figure 1 should report and visualize heritability values from all methods used, and visualize the correlations between them. The text should describe these results, the methods used, the heritability values reported and their correlation.

We agree the differences between estimates from different methods needed further explanation. We thoroughly scrutinized the publications describing each method, together with input from a colleague who is an expert in quantitative genetics (Jason Wolf, University of Bath).

The differences in estimates are due to the way in which the genetic relatedness matrix (GRM) is calculated. Several methods use standardized GRMs, which standardize the variance across SNPs by accounting for allele frequencies (see manuscript methods for GEMMA approach, lines 685-713; *sommer* calculates the GRM following (VanRaden, 2008)). This approach assumes causative alleles are rare in the mapping population, as might be expected when mapping disease traits in humans or mapping in a sample from a natural population. As our mapping population was produced by intercrossing eight inbred strains, causative alleles are not expected to be rare, even for deleterious phenotypes. In this case, GEMMA recommends using a centered GRM for mapping (GEMMA manual 4.4.2), which we followed. However, when calculating heritability estimates, GEMMA always accounts for allele frequency differences – by multiplying the genetic variance component by a standardization factor to (Zhou et al., 2013, supplementary text). This is because the ‘chip heritability’ calculated in GEMMA only estimates the heritability that can be explained by *genotyped* SNPs (Zhou and Stephens, 2012) – in contrast to narrow-sense heritability (*h^2^*), which is the proportion of phenotypic variance explained by all additive genetic effects. Chip heritability is useful for genotype prediction, such as in animal breeding and predicting disease risk in humans. Typically chip heritability values are much lower than narrow-sense heritability estimates from other methods (e.g. for height in humans *h^2^* ~0.8, chip heritability is 0.41; Zhou et al., 2013). This explains why the heritability estimates from GEMMA (and other methods using GRMs adjusting for allele frequencies) were lower than estimates from lme4QTL (narrow-sense heritability).

We are interested in estimates of narrow-sense heritability, which is the upper bound of what could be explained if all additive effects are known. However, we decided to report both chip heritability and *h^2^* in the manuscript (Results lines 143 – 161, Figure 1A-B, Supplementary File 1), for ease of comparison with some previous studies, which report estimates of chip heritability.

It is not possible to include categorical covariates in GEMMA, therefore we used lme4QTL to calculate both chip heritability and *h^2^* (Methods, lines 685-708), after first verifying that using the standardized GRM without covariates in lme4QTL produces the same results as GEMMA.

In addition, we added a comparison of *h^2^* estimates to PVE by significant SNPs in Results lines 279 – 287, Figure 1A-B.

There should be a clear discussion about the possible reasons for the high heritability values (and why they are lower using other methods).

We added an explanation of the difference between chip heritability and narrow-sense heritability estimates to the Discussion (lines 461-468).

I would also suggest including the analysis comparing heritability values across studies in the text, and include a visualization of these correlations, rather than just reporting the p-value.

We agree. We added a plot (Figure 1—figure supplement 5) showing the correlation of the narrow-sense heritability estimates of this study (lme4QTL) with Org et al., 2015 and Turpin et al., 2016. Supplementary File 1 includes heritability estimates from previous studies, along with our estimates.

Regarding code availability, I want to thank the authors for enhancing the README file on the github repository, which now provides a nice description of the pipeline and analysis steps. However, I am not sure if this is sufficient for readers who want to reproduce the results: looking at the code itself, it seems like there are commands to load scripts that are not included in the repository (e.g. the snp_heritability_lme4qtl.R script loads the script function_for_gemma.r that I couldn't find anywhere), and these scripts might not be able to be run on other machines. I recommend amending the github repository and scripts so that anyone who wishes to do so is able to run the analysis and reproduce the results.

Thank you for spotting that this script was missing. It has now been added. We however respectfully maintain that further development of our documented pipeline would amount to the creation of a new analysis package, which we feel would go beyond the scope of the current study. The github repository was rather intended as a basis for researchers to adapt these approaches to their individual datasets.

References:

VanRaden, PM (2008), ‘Efficient methods to compute genomic predictions.’, *J Dairy Sci*, 91 (11), 4414-23.

Zhou, Xiang and Matthew Stephens (2012), ‘Genome-wide efficient mixed-model analysis for association studies’, *Nat. Genet.*, 44 (7), 821-24.

Zhou, Xiang, Peter Carbonetto, and Matthew Stephens (2013), ‘Polygenic modeling with Bayesian sparse linear mixed models’, *PLoS Genet.*, 9 (2), e1003264.